# *Gas1* regulates embryonic tongue muscle proliferation, differentiation and maturation via alternative pathways to Hedgehog signaling

Gabrielle C. Audu, Sally Y. Rohan and Archana Kumari*

## ABSTRACT

Hedgehog (HH) signaling supports tongue and taste organ development. While the tongue is highly muscular, the role of HH signaling in muscle growth remains poorly understood. We recently showed the expression of HH receptor *Gas1* in postnatal lingual muscle. To understand the role of *Gas1* in the embryonic tongue, we first examined its expression using *Gas1^lacZ* mouse and GAS1 immunostaining. Our results reveal parallel gene and protein expression in epithelial taste buds, stroma and muscles. We assessed *Gas1* constitutive and muscle-specific conditional (E12.5-E18.5) gene deletion effects at E18.5. Constitutive *Gas1* deletion disrupts myoblast count, cell proliferation, differentiation, maturation and motor structures, and differentially affects the size and number of intrinsic tongue muscles. We unmask the expression of other HH co-receptors, CDON and BOC, in lingual epithelium, stroma or muscles, which, along with HH-responding GLI1 cells, persists, despite *Gas1* deletion. We propose an interplay of *Gas1* in distinct lingual compartments for tongue myogenesis, which is independent of HH signaling. We also suggest that while the cell-intrinsic roles of *Gas1* in muscle development may be redundant with other HH co-receptors, its cross-compartmental function is not.

KEY WORDS: Tongue myogenesis, Intrinsic lingual muscles, Hedgehog signaling, *Gas1*, BOC, CDON

## INTRODUCTION

The mammalian tongue is highly muscular and plays essential roles beyond detecting taste, including in swallowing and speech mechanisms (Su et al., 2020). In mouse, tongue muscle development begins around embryonic day (E) 10.5 with the formation of lingual swellings (Fig. 1A) (Zhang et al., 2022; Parada et al., 2012). These swellings consist of an epithelium and underlying mesenchyme (Parada et al., 2012). At E11.5, muscle progenitors begin migrating into these swellings. The swellings then fuse, forming the front two-thirds of the tongue at E12.5. This initial period is categorized as embryonic myogenesis, while the subsequent fetal myogenesis occurs between E14.5 and E17.5 (Fig. 1A) (Biressi et al., 2007; Stockdale, 1992). The

Department of Neuroscience, Rowan-Virtua School of Osteopathic Medicine, Virtua Health College of Medicine and Life Sciences of Rowan University, Stratford, NJ 08084, USA.

*Author for correspondence (kumari@rowan.edu)

 A.K., 0000-0001-5675-1389

embryonic phase establishes the foundational muscle structure (Murphy and Kardon, 2011; Hutcheson et al., 2009; White et al., 2010). During the fetal phase, myoblasts undergo rapid proliferation, fusion and differentiation, contributing to the increase in muscle mass and the refinement of muscle fiber architecture, which are vital for the muscle function postnatally. Lingual myofibers become discernible from E13.5 (Xu et al., 2022). They comprise both intrinsic and extrinsic muscles (Noden and Francis-West, 2006), though our focus here is on intrinsic muscles.

Intrinsic tongue muscles consist of four types of fibers (Fig. 1B): longitudinal, which runs along the length of the tongue from the tip to the back and includes the superior longitudinal (SLm) below the dorsal epithelium and the inferior longitudinal (ILm) above the ventral epithelium; vertical (Vm), which originates from the bottom of the tongue and courses ventrally towards the SLm; and transverse (Tm), which runs from the median septum to the lateral regions. These intrinsic muscles serve distinct roles in tongue movements, such as contraction (SLm and ILm), elongation (Vm) and flattening (Tm) (Sanders and Mu, 2013). Despite extensive studies on tongue muscle anatomy, myogenesis and function (Cobourne et al., 2019; Parada et al., 2012; Parada and Chai, 2015), the molecular mechanisms regulating embryonic tongue muscle differentiation and maturation have not been clearly elucidated.

Hedgehog (HH) signaling is essential for embryonic myogenesis of skeletal muscles, including tongue (Gustafsson et al., 2002; Munsterberg et al., 1995; Okuhara et al., 2019; Xu et al., 2022). The HH pathway initiates with the HH ligand binding to its membrane receptor, PTCH1, which modulates another membrane receptor, SMO, to regulate the transcription of target genes (Briscoe and Therond, 2013). Apart from PTCH1, three other membrane co-receptors, GAS1, CDON and BOC, can bind to HH ligand (Allen et al., 2007; Martinelli and Fan, 2007). Dual lipidated HH ligand is recruited by CDON and/or BOC, and sequentially transferred to GAS1 to remove lipids that allow HH-PTCH1 binding (Huang et al., 2022; Wierbowski et al., 2020) (Fig. 1C). These co-receptors can establish the ligand gradient in various tissues and developmental stages (Allen et al., 2011; Cobourne et al., 2004; Echevarría-Andino et al., 2023; Holtz et al., 2013; Izzi et al., 2011; Lee and Fan, 2001; Tenzen et al., 2006), but remain underexplored in tongue. HH signaling regulates myoblast migration and myofiber patterning of lingual muscle (Okuhara et al., 2019; Xu et al., 2022); whether HH signaling or its components play an equally essential role once lingual muscles are organized, i.e. during fetal myogenesis, has not yet been investigated.

Recently, we mapped HH ligand, receptors and transcription factors at different postnatal stages of the tongue (Kumari et al., 2024). Intriguingly, *Gas1* is the only component expressed in early postnatal tongue muscles. Further, *Shh*- and HH-responsive *Gli1* cells are not expressed in lingual muscles in any of the postnatal

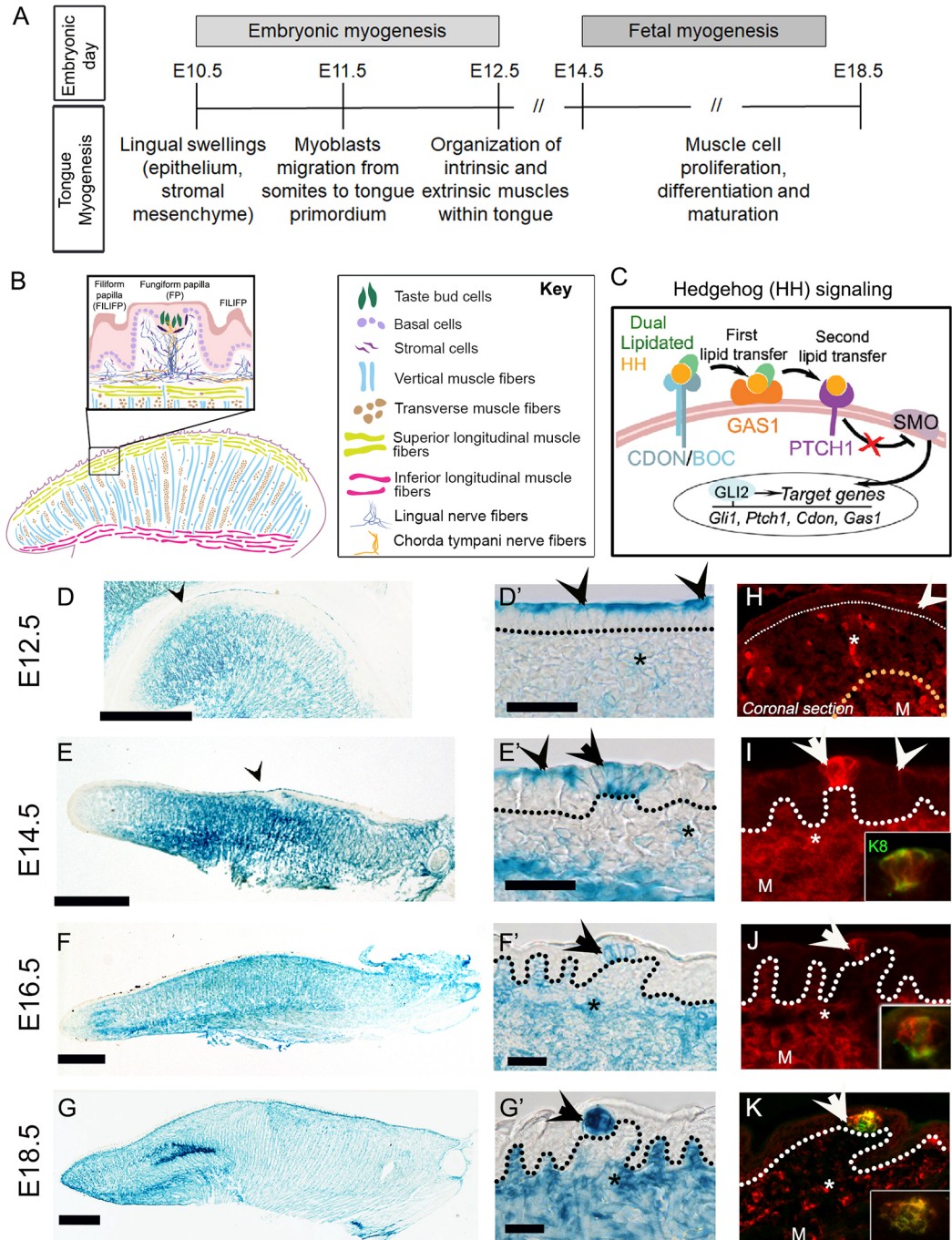

**Fig. 1. Gas1 expression in the embryonic tongue.** (A) Tongue development stages in the context of muscle maturation. (B) Diagram of sagittal tongue section illustrating intrinsic muscles. Inset indicates fungiform and filiform papillae, and underlying tissues. The key includes cell/fiber types. (C) The HH co-receptors CDON/BOC and GAS1 move the dual lipidated HH ligand to the PTCH1 receptor, thereby removing SMO receptor inhibition to activate target gene expression. (D-G) X-gal staining of *Gas1^lacZ^* sagittal tongue sections indicates expression in epithelium (arrowheads), stroma and muscles at E12.5 (D), E14.5 (E), E16.5 (F) and E18.5 (G). (D′-G′) Magnified images of epithelium and fungiform papillae demonstrating *Gas1^lacZ^* expression in the apical layer (arrowheads) at E12.5 (D′) and E14.5 (E′), in immature taste buds (arrows) between E14.5 and E18.5 (E′-G′), and in underlying stroma (asterisks) at all timepoints (D′-G′). (H-K) GAS1 immunostaining reveals protein expression similar to gene expression in the apical layer of the epithelium (arrowheads) at E12.5 (H) and E14.5 (I), in immature taste bud (K8, green, insets) between E14.5 and E18.5 (I-K, arrows), and in stroma (asterisks) and muscle cells (M) at all timepoints (H-K). Black and white dotted lines outline the epithelium; the yellow dotted line (H) outlines muscles. Scale bars: 500 μm in D-G; 25 μm in D′-G′ (also applying to H-K, respectively). Insets in I-K indicate taste bud primordium with GAS1 (red) and K8 (green) co-expression.

stages (Kumari et al., 2024), indicating that HH signaling may not directly influence muscle maturation. This raises an intriguing question: what role does *Gas1* play in tongue muscles?

Embryonic *Gas1* in the lingual mesenchyme was shown at E11.5 (Okuhara et al., 2019). However, data are not available beyond that

stage, nor about its role in muscle development. Together with *Boc* deletion, *Gas1* double mutants caused clefting of the pharyngeal tongue (Seppala et al., 2014). Separately, *in vitro* studies on hindlimb muscles suggest that *Gas1* promotes muscle differentiation, and a potential link with *Cdon* and *Boc* was suggested (Leem et al., 2011). Whether this

also occurs *in vivo* or in the tongue remains unknown. Together, there is a knowledge gap in the molecular mechanism of embryonic tongue muscle maturation, as well as its regulation by HH co-receptors.

Here, we have mapped lingual *Gas1* expression during embryonic development and characterized its roles at E18.5. Using a *lacZ* reporter mouse model and GAS1 antibody, we observed both gene and protein expression in the tongue taste bud primordium, stromal cells and muscle fibers throughout embryonic development. We employed constitutive whole-body and conditional (E12.5 onwards) muscle-specific *Gas1* gene deletion mouse models to investigate its functions in different tissue compartments. The loss of global *Gas1* affected tongue epithelium differentiation without affecting the phenotype. The stroma remained intact in both gene deletion models. Importantly, *Gas1* constitutive deletion significantly disrupted the lingual myoblast count, muscle cell proliferation, differentiation, maturation and functional motor structures, as well as SLm, ILm and Vm fiber size, count or relative frequencies. These significant changes were not observed in muscle-specific *Gas1* deletion, suggesting that *Gas1* in different tongue compartments contributes to the interplay necessary for muscle differentiation and maturation. Further, we demonstrated expression of other HH co-receptors, CDON and BOC, in lingual epithelium, stroma or muscles, and propose redundancy in their muscle-specific function.

## RESULTS

### *Gas1* gene and protein expression in embryonic tongue

We studied *Gas1* expression using *Gas1^lacZ^*, a knock-in reporter mouse model (Martinelli and Fan, 2007), at E12.5, E14.5, E16.5 and E18.5 (Fig. 1D-G′). X-gal staining of *Gas1^lacZ^* tongue sagittal sections revealed its extensive expression in lingual muscles from E12.5 through E18.5 (Fig. 1D-G). In addition, *Gas1^lacZ^* was expressed in the entire apical layer (but not at the tip) of the epithelium at E12.5 (Fig. 1D,D′, arrowheads) and E14.5 (Fig. 1E,E′, arrowheads). At E14.5, *Gas1^lacZ^* was further expressed in the fungiform papilla taste bud primordium (Fig. 1E′, arrow). While *Gas1^lacZ^* expression in the immature taste bud at E16.5 (Fig. 1F′, arrow) and E18.5 (Fig. 1G′, arrow) was maintained, it was not observed in the respective apical epithelium (Fig. 1F′,G′). Additionally, *Gas1^lacZ^* expression in underlying stroma increases overtime (Fig. 1D′-G′, asterisks). At all timepoints, X-gal staining of *lacZ*-negative tongues (control) elicited no blue product (Fig. S1A).

An equivalent expression pattern was observed in the apical layer of epithelium with GAS1 antibody staining at E12.5 (coronal section, Fig. 1H, arrowhead) and E14.5 (Fig. 1I, arrowhead). Similarly, GAS1 was observed in the K8^+ taste bud primordium from E14.5 onwards (Fig. 1I-K, arrows, insets), and in the stroma (asterisks) and muscle (M) cells from E12.5-E18.5 (Fig. 1H-K). Together, the data suggest that GAS1 protein expression mimics *Gas1* gene expression in tongue apical epithelium, immature taste bud, stromal cells and muscle fibers throughout embryonic development.

### *Gas1* loss of function alters tongue shape

We utilized the *Gas1^lacZ/lacZ^* homozygous mouse to study *Gas1* loss-of-function. X-gal staining of mutant tongues from E12.5 through E18.5 confirmed efficient global *Gas1* deletion (Fig. 2A). Mutant (*Gas1^lacZ/lacZ^*) embryos were smaller compared to control (*Gas1^+/+^*) and heterozygous (het, *Gas1^lacZ/+^*) embryos at E12.5 (Fig. 2B), and the effects were more pronounced at E18.5 (Fig. S1B). Further, head length was significantly reduced in E18.5 mutant compared to control and/or het (Fig. 2C,C′), corroborating previous findings (Seppala et al., 2007). We restricted all our data analyses to E18.5 from this point forward.

Jaw angle (yellow dotted lines) and tongue width (black dotted lines) were significantly smaller in mutant than control/het (Fig. 2D-D″). We then measured the tongue length from the tip to the base (Fig. 2E, control, dotted line) and height at three distinct locations – tip, body and posterior (arrows). Length was significantly reduced in mutant compared to control and het (Fig. 2E,E′). However, the height of the tongue tip, body and posterior in mutant did not change significantly compared to control and het (Fig. 2E,E″). These findings suggest that *Gas1* plays a crucial role in defining tongue size and shape.

### *Gas1* deficiency alters lingual epithelial cell differentiation

We investigated the effects of *Gas1* loss on tongue taste fungiform and non-taste filiform papillae (Fig. 1B) in control, het and mutant groups. Qualitative histological analysis revealed no apparent change in the morphology of lingual papillae in any of the experimental groups (Fig. 3A-C). Further, fungiform papilla density (Fig. S1C), percentage of typical fungiform papilla (Fig. S1D) and taste bud area (Fig. S1E) also remained unaltered.

Within lingual epithelium (Fig. 3D, Ecad), Ki67^+ proliferating cells are expressed in fungiform papilla basal epithelial (arrowheads) and perigemmal cells (arrows). The number of Ki67^+ cells per fungiform papilla was maintained in mutant compared to control or het (Fig. 3D-F, Fig. S1F). Similarly, Ki67^+ cells were maintained in the filiform papilla basal epithelium in all groups (Fig. 3D-F, insets, arrowheads).

Lingual basal epithelium, labeled with K5, is progenitor of taste bud and filiform papilla (Liu et al., 2013; Okubo et al., 2009). While control (Fig. 3G) and het (Fig. 3H) showed K5^+ cells in the basal layer of fungiform and filiform papillae (insets), mutant displayed a reduction in K5^+ cells in the lower half of both lingual papillae (Fig. 3I, arrows, inset).

As *Cdon* is specifically expressed in the lower half of postnatal lingual epithelium (Kumari et al., 2024), we investigated its expression in fungiform papilla. Antibody staining revealed CDON expression, similar to *Cdon* expression, in the lower half of fungiform papilla in control, het and mutant (Fig. 3G′-I′, arrows). Notably, K5 expression was eliminated only from CDON^+ cells in mutants (Fig. 3I,I′, arrows), yet these K5^−;CDON^+ basal cells retained proliferation in mutants (Fig. S1G, arrows).

Intriguingly, unlike postnatal stages (Kumari et al., 2024), we observed CDON^+ cells within embryonic taste buds in control (Fig. 3G′, arrowheads, inset). The CDON^+ taste cells were also observed in het (Fig. 3H′, arrowhead, inset) but not in mutant (Fig. 3I′).

To investigate whether GAS1 is an essential receptor for HH signaling transduction, we examined the effect of its loss on SHH distribution and HH-responding GLI1 activity in fungiform papilla. SHH expression in taste buds remained intact in all groups (Fig. 3J-L). *Gas1^lacZ^;Gli1^eGFP^* bi-transgenic animals were employed to analyze GLI1 cells. In control (Fig. 3M), we observed GLI1 (GFP^+) expression in fungiform papilla perigemmal cells (arrow), basal epithelium (arrowhead) and stromal cells (asterisk), as previously reported (Kumari et al., 2015, 2017). Importantly, HH signaling remained active in these regions in het and mutant (Fig. 3N,O, arrows, arrowheads, asterisks).

Overall, we observed a notable reduction in K5^+ lingual epithelial basal cells in *Gas1* mutants. Despite this reduction, the mutants maintained epithelial cell proliferation and thereby the structural integrity of fungiform and filiform papillae. Furthermore, the retained expression of SHH and GLI1^+ cells suggest that *Gas1^+^* taste bud cells are dispensable for SHH transport and paracrine HH signaling transduction in the fungiform papilla.

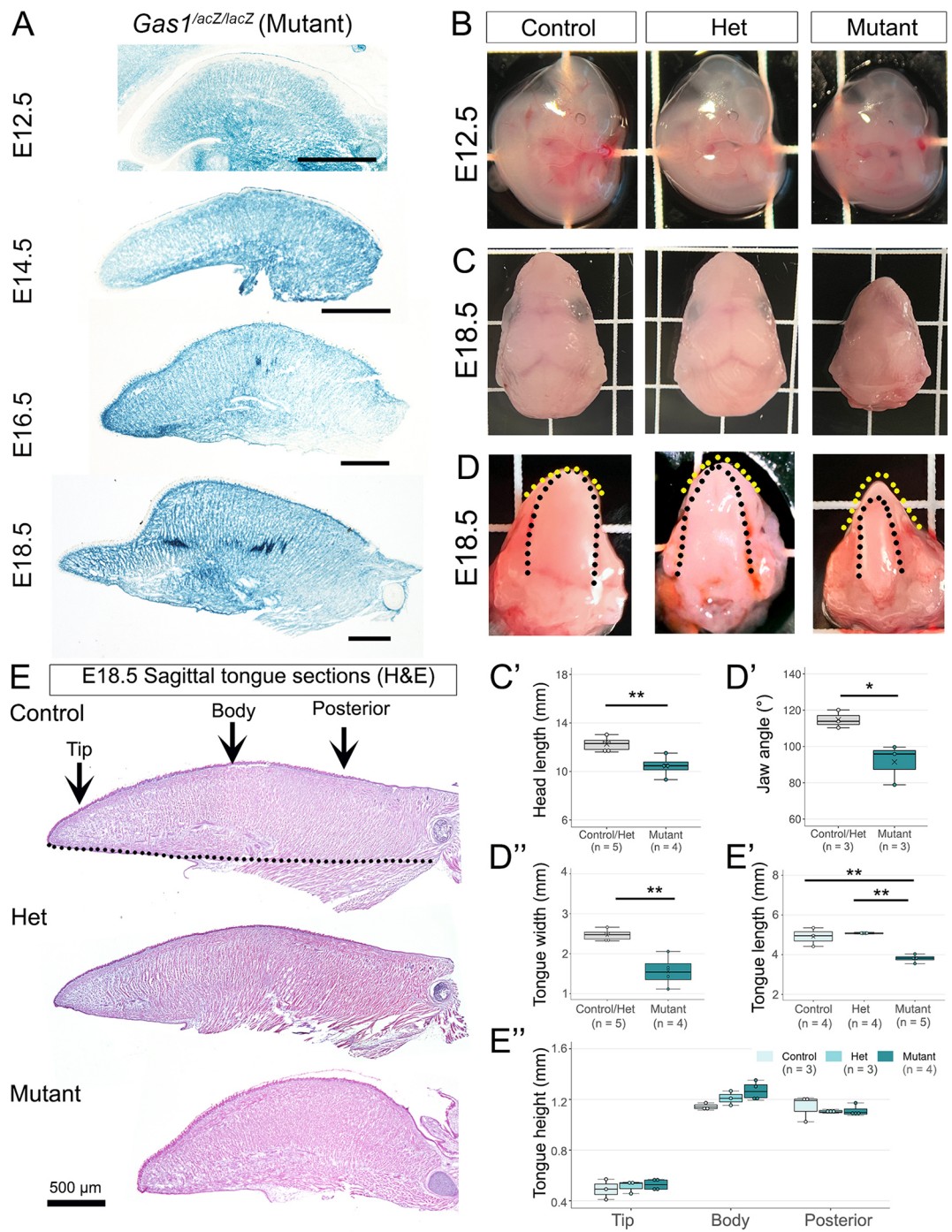

**Fig. 2. *Gas1* loss affects embryonic tongue development.** (A) X-gal staining of tongue at E12.5, E14.5, E16.5 and E18.5 in the *Gas1^{lacZ/lacZ}* mutant validates efficient gene deletion. Scale bars: 500 µm. (B) At E12.5, embryo size decreases in mutant (*Gas1^{lacZ/lacZ}*) compared to control (*Gas1^{+/+}*) and Het (*Gas1^{lacZ/+}*). (C-D″) At E18.5, head length (C,C′), jaw angle (D, yellow dotted lines, D′) and tongue width (D, black dotted lines, D″) are significantly disrupted in mutant compared to control/het. (E) H&E stained sagittal sections of control, het and mutant tongues at E18.5. Dotted line denotes tongue length; arrows indicate location of measurement for tongue height. Tongue length (E′) is significantly decreased in mutant compared with control or het. Tongue height at the tip, body and posterior (E″) remains similar in all groups. Numbers in parentheses indicate the number of tongues analyzed. In the box and whisker plots, the horizontal line represents the median, the whiskers represent the maximum and minimum values, and the box indicates the interquartile range. X indicates mean. *$P \leq 0.05$, **$P \leq 0.01$ (two-tailed *t*-test in C′-D″ or one-way ANOVA in E′-E″).

## Lingual connective tissue core and innervation are maintained after *Gas1* deletion

Underneath the lingual epithelium lie the stromal cells that secrete collagen, the main component of the connective tissue core (Hosokawa et al., 2010). As *Gas1* is expressed in the stromal cells (Fig. 1D-K), we assessed expression of vimentin (marker of stromal cells) and collagen in control, het and mutant. In all groups, vimentin protein expression, as indicated by immunostaining (Fig. 4A) and gene expression using qPCR (Fig. S1H), was maintained. All the vimentin⁺ stromal cells were GLI1⁺ (Fig. 4B, Control), and the stromal GLI1 expression was maintained in het and mutant (Fig. 4B). Additionally, collagen expression in

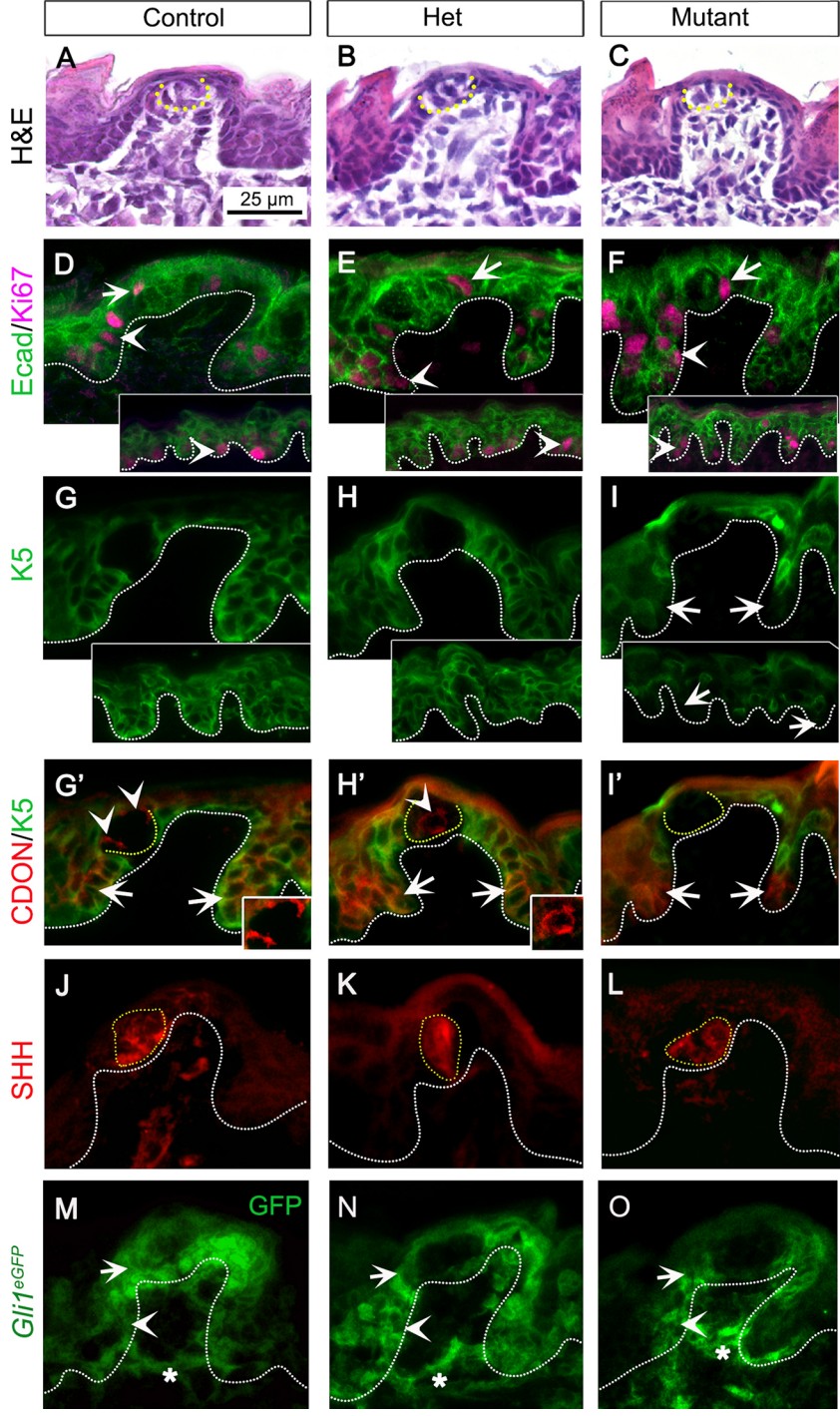

**Fig. 3. *Gas1* deletion disrupts lingual epithelium differentiation.** (A-C) H&E staining indicates no apparent changes in lingual papillae and taste buds in control (A), het (B) and mutant (C). (D-F) Ki67⁺ proliferating cells (purple) in epithelium (Ecad, green) are maintained in fungiform and filiform papillae (insets) basal cells (arrowheads) and in fungiform papilla perigemmal cells (arrows) in control (D), het (E) and mutant (F). (G-I) Undifferentiated K5⁺ cells (green) in fungiform papilla and filiform papillae (insets) basal epithelium in control (G), het (H) and mutant (I). Arrows indicate K5 loss in the lower half of the epithelium in mutant (I). (G′-I′) CDON (red) is expressed in the lower half of fungiform papilla basal epithelium (arrows) and within taste bud (arrowheads, insets) in control (G′) and het (H′). In mutant (I′), CDON expression is maintained in epithelium but lost in taste bud (arrows). Co-expression with K5 (green) indicates loss of K5 in CDON⁺ cells in mutant (I′). (J-L) SHH (red) expression within taste bud is retained in control (J), het (K) and mutant (L). (M-O) GFP immunostaining in *Gas1^lacZ^;Gli1^eGFP^* tongue (green) demonstrates GLI1 expression in fungiform papilla perigemmal cells (arrows), basal epithelium (arrowhead) and stromal cells (asterisks) in control (M), het (N) and mutant (O). Scale bar in A applies to all images. White and yellow dotted lines outline the epithelium and taste bud, respectively.

the tongue extracellular matrix was similar in all groups (Fig. 4C, blue).

Fungiform papilla core is innervated by somatosensory lingual nerve while the taste buds are innervated by multimodal chorda tympani nerve (Fig. 1B) (Kumari and Mistretta, 2023). To test whether *Gas1⁺* cells within taste bud primordium and fungiform papilla connective tissue core have a nerve guidance role, we qualitatively studied innervation using three markers. β-Tubulin labels neuronal microtubules non-selectively (Donnelly et al., 2022). In all groups, β-tubulin expression pattern and density within K8⁺ taste bud primordium (Fig. 4D, asterisk), fungiform papilla core (arrowhead) and along apical epithelium (arrow) was

retained. P2X3, a selective marker of chorda tympani nerve fibers, was observed in both intragemmal (Fig. 4E, K8⁺ taste bud primordium, asterisk) and extragemmal nerve fibers at perigemmal sites (arrow) coursing through fungiform papilla core (arrowhead) across all experimental groups. Neurofilament-heavy (NF) is a selective marker for myelinated lingual nerves. In all groups, NF fiber (Fig. 4F) organization and pattern remained consistent in the core of the fungiform papillae without entering the taste bud. These data suggest that the structural integrity, organization and distribution of embryonic tongue stromal cells, nerve fibers and stromal HH signaling are unaffected by global *Gas1* deletion.

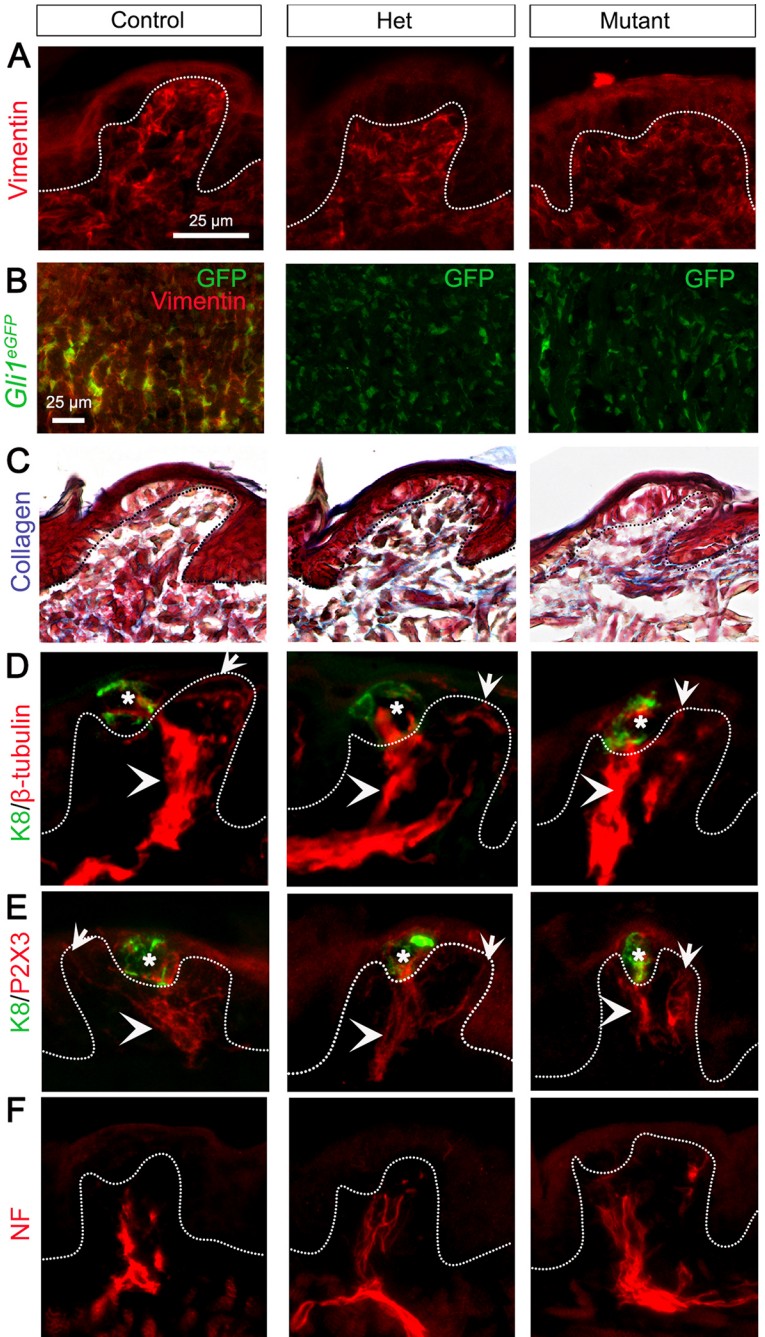

**Fig. 4. Fungiform papilla stromal cells and innervation are retained in the *Gas1* mutant.** (A) Vimentin⁺ stromal cells (red) are maintained in control, het and mutant. (B) GLI1 (GFP, green) is co-expressed with vimentin (red, control) and is maintained in het and mutant. (C) Masson's trichrome staining indicates retained collagen expression (blue) across all groups. (D,E) β-Tubulin⁺ neuronal microtubules (D, red) and P2X3⁺ chorda tympani fibers (E, red) are expressed in taste bud (K8, green, asterisks) and fungiform papilla core (arrowheads), and run along the apical epithelium (arrows) in all experimental groups. (F) NF⁺ lingual nerve fibers (red) are labeled in fungiform papilla core across all experimental groups. Scale bar in A applies to all images except B. Dotted lines outline the epithelium.

### *Gas1* mutants exhibit disruptions in lingual myoblast count, muscle cell proliferation, differentiation, maturation and neuromuscular junctions

Various markers that label different myogenic stages (Fig. 5A, blue) were quantified in control, het and mutant tongues. We observed a significant increase in the number of Ki67⁺ proliferating muscle cells (Fig. 5B,B′) and Pax7⁺ myoblasts (Fig. 5C,C′) in mutant compared to control and het. However, muscle cell fusion, as indicated by desmin, remained intact across experimental groups (Fig. S1I). In contrast, there was significantly decreased myogenin⁺ differentiating myocytes (Fig. 5D,D′) and consequently reduced mature myofibers, as evidenced by myosin heavy chain (MyHC) staining (Fig. 5E,E′) in mutant compared to control and het. Further, MyHC-embryonic, a regulator of myoblast differentiation (Agarwal

et al., 2020) was significantly reduced in mutant compared to control and het (Fig. 5F,F′). MyHC-slow was also substantially reduced in mutant (Fig. S1J,J′), whereas MyHC-neonatal remained unchanged (Fig. S1K,K′)

The neuromuscular junction (NMJ) or motor endplate (Fig. 5A, boxes) is formed when mature myofibers establish synapses with motor nerve fibers (green). Presynaptic nerves at the NMJ release the neurotransmitter acetylcholine (ACh), which binds to post-synaptic ACh receptors (AChRs) located on the muscle endplate, facilitating muscle contractions (Engel et al., 2015). We quantified the number and size of AChRs labeled using α-bungarotoxin staining (Fig. 5G, red). The number of AChR⁺ endplates showed an increasing trend in mutant compared to control and het (Fig. S2A). However, the area occupied by individual endplates

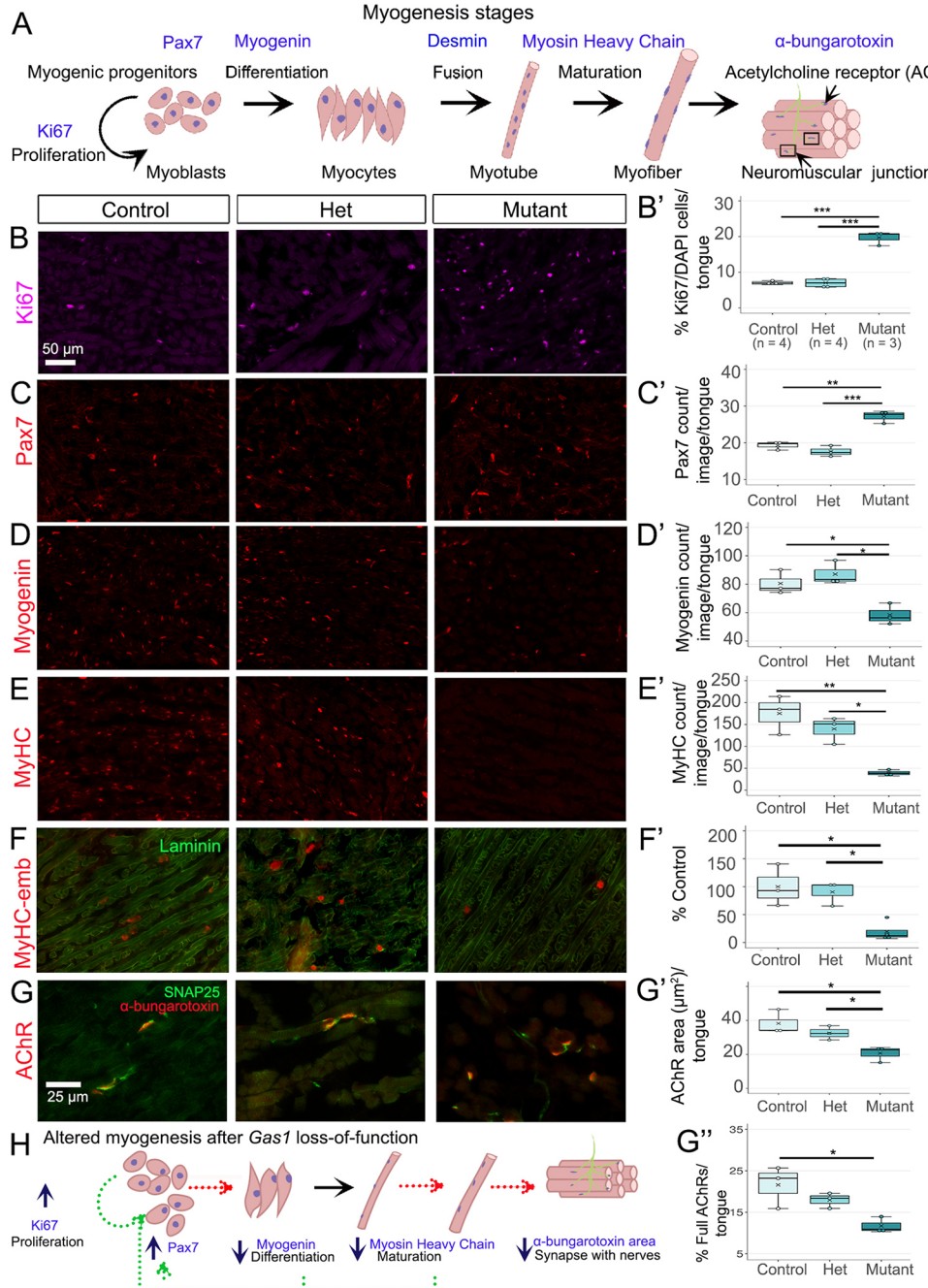

**Fig. 5. *Gas1* loss affects fetal myogenesis.** (A) Diagram of the myogenesis stages and their markers (blue). (B-C′) Ki67⁺ proliferating cells (B, purple, B′) and Pax7⁺ myoblasts (C, red, C′) are significantly increased in mutant compared to control and het. (D-F′) Myogenin⁺ differentiating muscle fibers (D, red, D′), MyHC⁺ mature fibers (E, red, E′) and MyHC-embryonic (F, red, F′; laminin, green) are significantly decreased in mutant compared to control and het. (G-G″) Immunostaining of presynaptic nerves (SNAP25, green) followed by postsynaptic acetylcholine receptor (AChRs) staining by α-bungarotoxin (red) in control, het and mutant (G). The AChR area is significantly smaller in mutant than control (G′). Full AChR (>80% overlap between SNAP25 and α-bungarotoxin) significantly declined in mutant compared to control (G″). Scale bar in B applies to C-E. In the box and whisker plots, the horizontal line represents the median, the whiskers represent the maximum and minimum values, and the box indicates the interquartile range. X indicates the mean. *$P \leq 0.05$, **$P \leq 0.01$, ***$P \leq 0.001$ (one-way ANOVA with Tukey's HSD post-hoc test). Number of tongues (*n*) analyzed is 3 unless noted otherwise. (H) Schematic representation of the disruptions in myogenesis after *Gas1* loss.

was significantly smaller in mutant relative to control (Fig. 5G′). To quantify the percentage of innervated endplates, we co-stained for α-bungarotoxin and the presynaptic nerve marker SNAP25 (Fig. 5G green). We observed a significant reduction in endplates fully occupied by SNAP25 fibers (>80% overlap) in mutant compared with control (Fig. 5G″). The number of axonal inputs was substantially reduced in mutant compared to control and het (Fig. S2B). Importantly, the extent of presynaptic invasion showed no correlation with endplate area in any of the experimental groups (Fig. S2C), and the percentage of vacant AChRs lacking SNAP25⁺ fibers was comparable across groups (Fig. S2D).

Together, *Gas1* deletion led to significantly reduced lingual muscle differentiation, maturation, AChR area and innervation (Fig. 5H, red arrows). The loss of mature myofibers potentially triggered a compensatory mechanism in muscle tissue, where increased Pax7⁺ myoblast proliferation attempted to offset the deficit in mature muscle fibers (Fig. 5H, green arrows). The data also suggest that fetal myogenesis (Fig. 1A) is altered in *Gas1* mutants, while myoblast migration (desmin⁺ cells) during the embryonic phase remain unaffected.

### *Gas1* loss of function differentially affects lingual intrinsic muscle fiber count and size

Given our findings that tongue muscle maturation is disrupted after *Gas1* deletion, we individually investigated all four intrinsic muscles for count and cross-sectional area (CSA) (Kletzien et al., 2018; Smith and Barton, 2014), followed by categorizing the fibers by size to determine their CSA relative frequencies. As the lingual intrinsic muscle fibers run in different directions (Fig. 1B), we sectioned the tongue in different planes for quantification

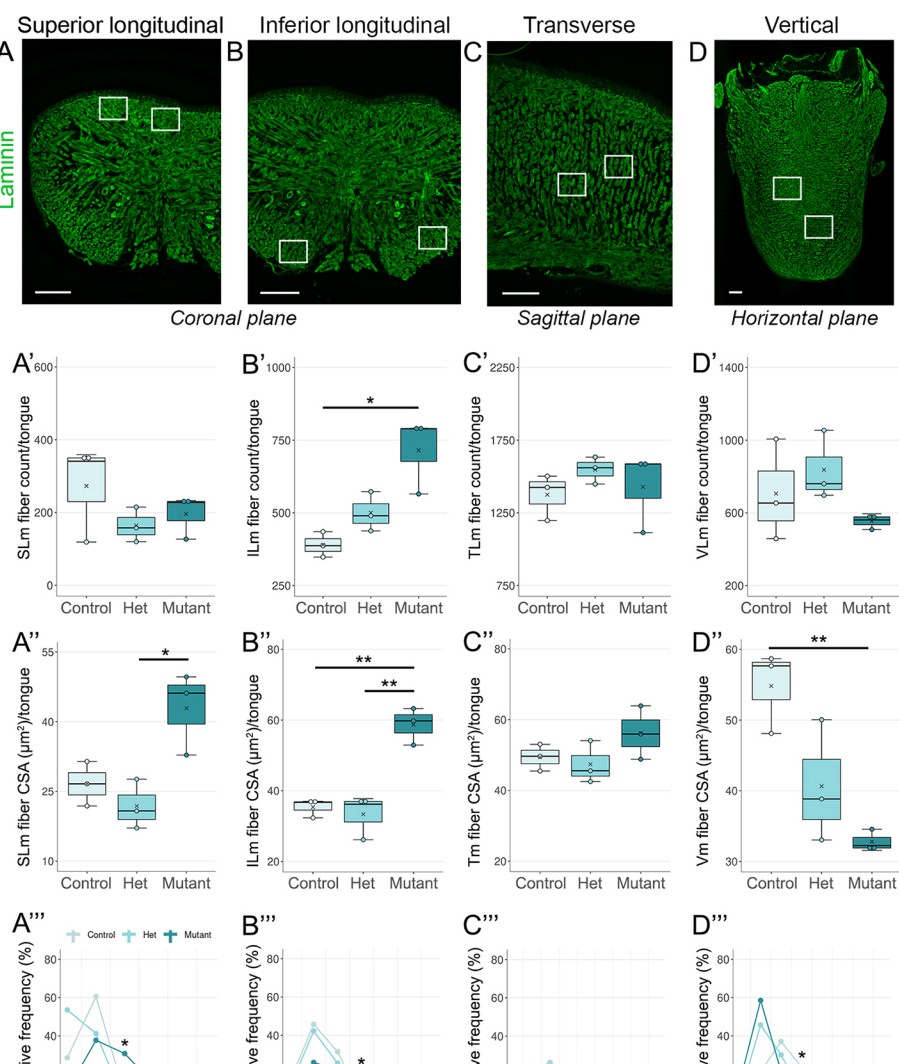

**Fig. 6. _Gas1_ deletion differentially affects lingual intrinsic muscle types.** (A-D) Longitudinal muscles, superior (SLm) (A) and inferior (ILm) (B), transverse muscle (Tm) (C) and vertical muscle (Vm) (D) in coronal, sagittal and horizontal sections, respectively, are outlined with laminin. Rectangles indicate measurement locations. (A′-A‴) SLm fiber count does not change between groups (A′). SLm fiber cross-sectional area (CSA) increased in the mutant compared to control and het (A″). SLm fiber size relative frequency is altered in mutant (A‴). (B′-B‴) ILm fiber count (B′) and CSA (B″) are significantly increased in mutant compared to control and/or het. The size distribution of mutant ILm fibers is altered (B‴). (C′-C‴) Tm fiber count (C′), CSA (C″) and size distribution (C‴) remain consistent in all groups. (D′-D‴) Vm fiber count is similar in all groups (D′). Vm fiber CSA is significantly reduced in mutant compared to control (D″). The fiber size distribution of Vm shifted towards the left in the mutant compared to control and het (D‴). Scale bars: 200 μm. In the box and whisker plots, the horizontal line represents the median, the whiskers represent the maximum and minimum values, and the box indicates the interquartile range. Line graphs show the mean. *$P \leq 0.05$, **$P \leq 0.01$ (one-way ANOVA with Tukey's HSD post-hoc test). Number of tongues analyzed is 3 in all groups.

(Fig. 6A-D): coronal (SLm and ILm), sagittal (Tm) and horizontal (Vm). We used laminin immunostaining to identify the muscle fibers, and the regions selected for study are indicated in Fig. 6A-D.

The number of SLm fibers showed no significant variation across experimental groups (Fig. 6A′), but their CSA was significantly higher in mutant compared to het (Fig. 6A″). The highest proportion of mutant and control fibers measured 21-40 μm², but the frequency was ~40% in mutant compared to ~60% in control. Further, in mutant, SLm fibers with a CSA between 41 and 60 μm² were significantly increased in comparison to control (Fig. 6A‴).

Both the number of ILm fibers (Fig. 6B′) and their overall CSA (Fig. 6B″) were significantly higher in mutant than those of control and/or het. While most fibers measured 21-40 μm², their frequency was ~30% in mutant, compared to ~50% in control or het (Fig. 6B‴). Additionally, ILm fibers with a CSA between 61 and 80 μm² were significantly increased in mutant compared with control (Fig. 6B‴).

The Tm fiber count (Fig. 6C′) and overall CSA (Fig. 6C″) remained the same across all experimental groups. There was a similar CSA distribution of Tm fibers in all groups, where the

highest proportion of fibers (~25%) was in the size range of 41-60 μm² (Fig. 6C‴).

The number of Vm fibers (Fig. 6D′) was similar across all groups, whereas the overall CSA (Fig. 6D″) was significantly reduced in mutant compared to control. The highest frequency of Vm fibers in control (~37%) had a CSA interval between 41 and 60 μm², while more than 55% of fibers in mutant were 21-40 μm² (Fig. 6D‴). In addition, the CSA frequency of fibers between 61-80 μm² and 101-120 μm² was significantly reduced in mutant compared to control (Fig. 6D‴).

In summary, although _Gas1_ is expressed in all tongue muscles, its deletion had varied effects on each intrinsic muscle. ILm exhibited changes in both cell count and size, whereas SLm and Vm were affected significantly in size. Notably, Tm remained unaffected.

### Muscle-specific _Gas1_ deletion does not disrupt the lingual epithelium, stroma or muscle

Constitutive deletion of _Gas1_ did not impair tongue or muscle formation by E12.5 (Fig. 2A), suggesting that _Gas1_ is not essential for myoblast migration or muscle patterning during embryonic

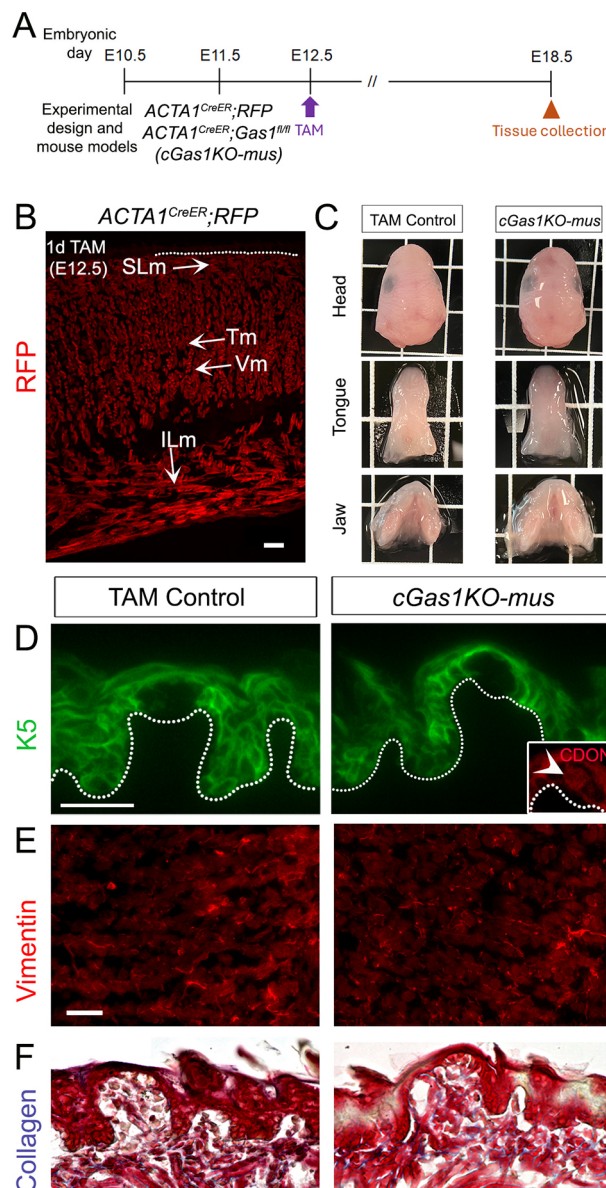

**Fig. 7. Muscle-specific *Gas1* deletion does not disrupt tongue tissues.**
(A) Schematic showing experimental design. (B) RFP immunostaining in
*ACTA1^CreER^;RFP* mouse treated with tamoxifen (TAM) at E12.5 validates
expression in all intrinsic muscles (SLm, ILm, Tm and Vm) at E18.5.
(C) TAM control (cre/flox negative littermates) and *ACTA1^CreER^;Gas1^fl/fl^*
(*cGas1KO-mus*) display similarly sized and shaped heads, tongues and
jaws. (D) K5⁺ differentiating cells (green) in lingual papillae are retained in
TAM control and *cGas1KO-mus*. CDON (red, inset, arrowhead) within the
taste bud is maintained in *cGas1KO-mus*. (E,F) Vimentin (E, red) and
collagen (F, blue) expression is maintained in TAM control and *cGas1KO-
mus*. Dotted lines outline the epithelium. Scale bars: 25 µm.

myogenesis (Fig. 1A). Therefore, we investigated *Gas1* muscle-
specific role after E12.5 using skeletal muscle-specific CreER driver
*ACTA1^CreER^* (McCarthy et al., 2012) and *Gas1^fl/fl^* mouse model
(Li et al., 2019) (Fig. 7A). We used the *ACTA1^CreER^;RFP* reporter
mouse, which received one dose of tamoxifen (TAM) at E12.5, to
confirm RFP expression in all intrinsic muscle types at E18.5
(Fig. 7A,B). We then induced muscle-specific *Gas1* deletion using
*ACTA1^CreER^;Gas1^fl/fl^* bi-transgenic mouse from E12.5 to E18.5,
grouped as *cGas1KO-mus* (Fig. 7A). Cre or flox negative animals

served as TAM control. Phenotypically, the sizes of head, tongue
and jaw were not affected in *cGas1KO-mus* compared to TAM
control (Fig. 7C).

Unlike the whole-body *Gas1* deletion (Fig. 3I), K5 expression in
the lingual basal cells was maintained in both TAM Control and
*cGas1KO-mus* (Fig. 7D). Similarly, CDON expression within the
taste bud (Fig. 3G′, inset), which was eliminated in mutant
(Fig. 3I′), was maintained in the *cGas1KO-mus* (Fig. 7D, inset,
arrowhead). Further, expression of both vimentin and collagen
were retained in *cGas1KO-mus* and TAM control (Fig. 7E,F).
Together, these findings confirm that there is no *Gas1*-specific
muscle contribution to the epithelium or stroma in the embryonic
tongue.

In muscles, the number of Ki67⁺ proliferating cells (Fig. 8A,A′),
Pax7⁺ myoblasts (Fig. 8B,B′), myogenin⁺ differentiating fibers
(Fig. 8C,C′), MyHC⁺ mature fibers (Fig. 8D,D′) and
α-bungarotoxin⁺ AChR motor endplates (Fig. 8E,E′) were not
significantly different in *cGas1KO-mus* and TAM control. Thus,
*Gas1* might not have cell-intrinsic roles in regulating tongue fetal
myogenesis after E12.5, or the function might be redundant with
other HH co-receptors.

## HH co-receptors CDON and BOC may have overlapping roles with GAS1 in muscles

GAS1 was expressed in tongue stroma (s) and muscles (m)
(Fig. 9A). Although *Cdon^lacZ^* was not detected in postnatal tongue
muscle (Kumari et al., 2024), we demonstrate CDON expression in
the embryonic muscles (Fig. 9B, arrow). Using a BOC antibody, we
observed expression in lingual epithelium, stroma and muscles
(arrow) (Fig. 9C). Notably, similar to CDON (Figs 3G′ and 9B),
epithelial BOC expression was predominantly in the lower half of
the basal layer, as seen with K5 (red) co-expression (Fig. 9C inset,
arrowheads).

We investigated GAS1, CDON and BOC expression in *Gas1*
mutant (Fig. 9D-F) and *cGas1KO-mus* (Fig. 9G-I). GAS1 was
eliminated from all tissues in mutant (Fig. 9D), while stromal
expression was intact in *cGas1KO-mus* (Fig. 9G), further confirming
the validity of our mouse models. Importantly, in both mutant and
*cGas1KO-mus*, the expression of CDON (Fig. 9E,H) and BOC
(Fig. 9F,I) in muscles (arrows) and/or stroma did not change.

These data indicate potential for redundancy of CDON and BOC
with GAS1 in muscle.

## DISCUSSION

Similar to embryonic myoblasts, fetal myoblasts need precise
signaling regulations to coordinate their growth and maturation
(Chal and Pourquie, 2017). We recently demonstrated that the HH
co-receptor *Gas1* is present in postnatal lingual muscles (Kumari
et al., 2024). However, the biological relevance of this finding
remains to be investigated. A range of embryonic craniofacial
malformations have been noted following *Gas1* deletion (Allen
et al., 2007; Echevarría-Andino and Allen, 2020; Khonsari et al.,
2013; Martinelli and Fan, 2007; Seppala et al., 2007, 2014) but the
tongue has not been previously studied. Here, we reveal the
expression and vital regulatory function of *Gas1* in embryonic
tongue intrinsic muscle growth and maturation.

### GAS1 is a membrane-bound receptor protein throughout the embryonic tongue development

GAS1 is attached to the outer cell membrane via a
glycosylphosphatidylinositol anchor (Ruaro et al., 2000; Stebel
et al., 2000) but also exists as a secreted protein (Bautista et al., 2018;

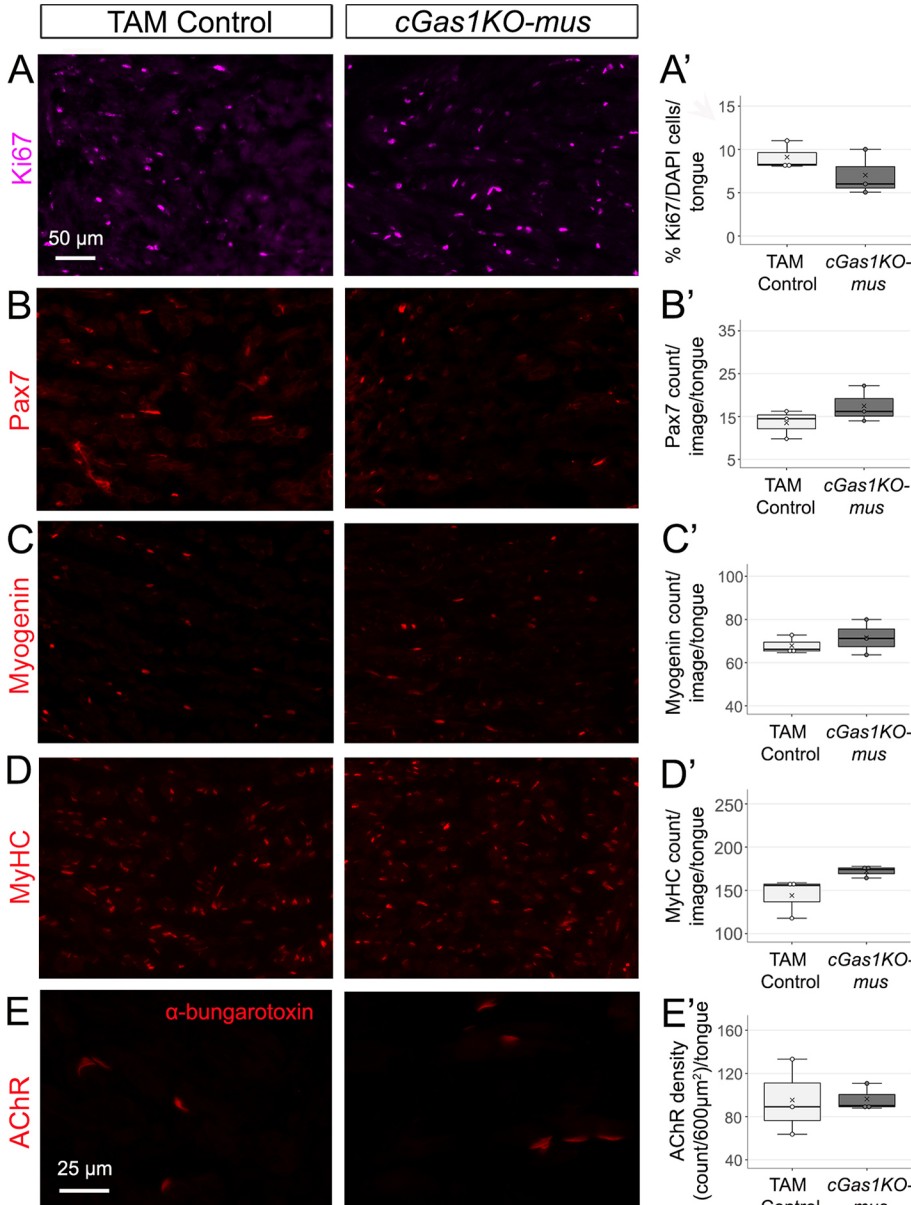

**Fig. 8. Muscle-specific *Gas1* deletion does not disrupt lingual muscles.** (A-E′) Muscle cell proliferation (Ki67, purple, A,A′), myoblast count (Pax7, red, B,B′), myofiber differentiation (myogenin, red, C,C′), maturation (MyHC, red, D,D′) and AChR$^+$ motor endplates (α-bungarotoxin, red, E,E′) are retained in TAM control and *cGas1KO-mus*. Scale bar in A applies to all images except E. In the box and whisker plots, the horizontal line represents the median, the whiskers represent the maximum and minimum values, and the box indicates the interquartile range. X indicates the mean. Number of tongues analyzed is 3 in all groups.

van Roeyen et al., 2013), suggesting that its functional roles may extend beyond membrane-bound signaling. In embryonic tongue, the expression of GAS1 protein completely aligns with the *Gas1* gene, indicating that GAS1 remains localized to the cells where the gene is active for non-secretory functions.

*Gas1* is expressed in embryonic tongue taste bud primordium, stroma and muscles, similar to postnatal stages (Kumari et al., 2024). However, unlike the postnatal expression decline in stroma and elimination in muscles (Kumari et al., 2024), *Gas1* expression remains robust in embryonic tongue stroma and muscles. This sustained expression suggests a continuous requirement for *Gas1* for embryonic tongue development, though functional validation is needed to confirm this role.

### Differential overlap of GAS1 with the other HH co-receptors CDON and BOC in embryonic tongue compartments

GAS1 mediates SHH ligand transfer from CDON and/or BOC to PTCH1 to induce HH signaling (Huang et al., 2022; Wierbowski et al., 2020) (Fig. 1C). CDON and BOC often overlap (Mulieri et al., 2002, 2000) and can function redundantly with GAS1 to promote HH signaling during ventral neural patterning (Allen et al., 2011) and craniofacial development (Pineda-Alvarez et al., 2012; Seppala et al., 2007; Zheng et al., 2010). However, their interactions can be antagonistic (Echevarría-Andino and Allen, 2020) or independent (Allen et al., 2011, 2007; Fabre et al., 2010; Okada et al., 2006). In embryonic tongue, GAS1 and CDON are observed in the fungiform papilla taste bud primordium (Figs 1K and 3G′). While CDON appears to be localized to a few taste bud cells, *Gas1* is expressed in all taste bud cells (Fig. 10A). Notably, *Gas1* is absent from the basal epithelium, where both CDON and BOC are expressed (Figs 9A-C and 10A). Further, lingual stromal cells express GAS1 and BOC but lack CDON expression (Figs 9A-C and 10A). Lingual muscles (Figs 9A-C and 10A) and myoblasts (Okuhara et al., 2019) express all three HH co-receptors but the extent of co-expression has not yet been investigated. The partial or non-overlapping expression patterns of GAS1 with CDON and/or BOC in the lingual epithelium, taste buds and stroma, along with co-expression in

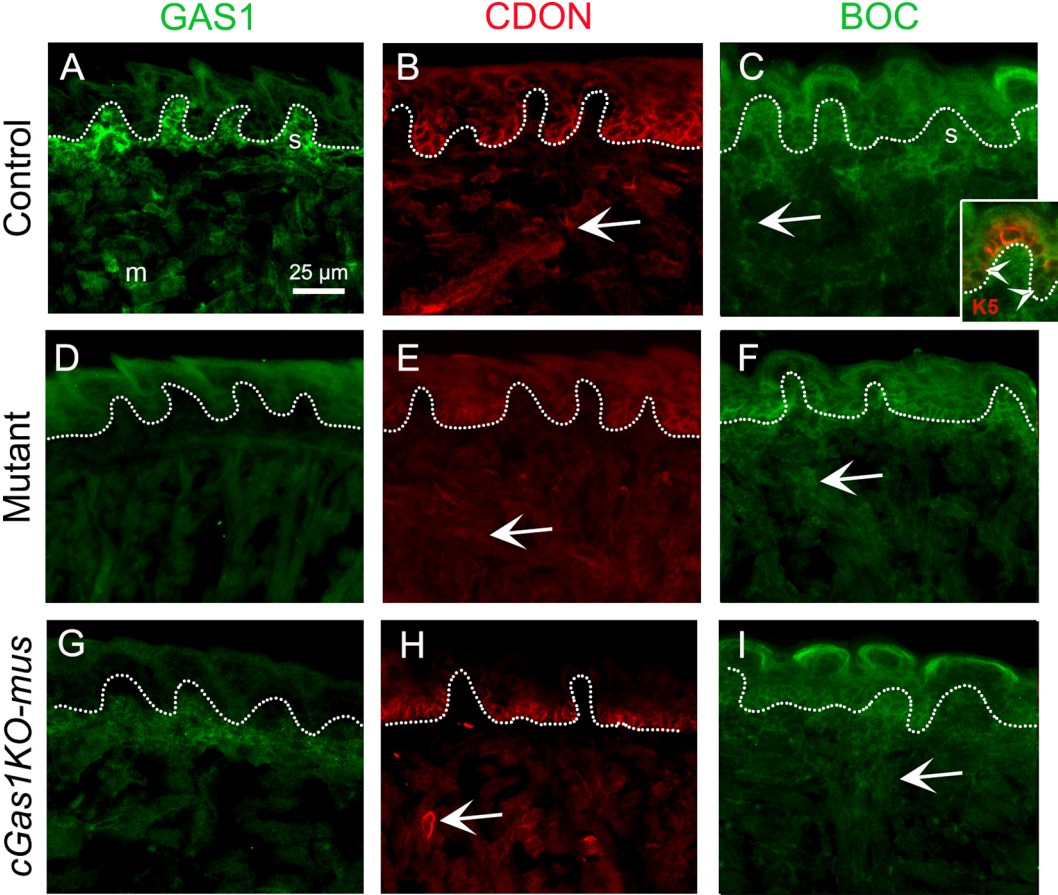

**Fig. 9. Differential expression of HH co-receptors CDON and BOC compared to GAS1 and their stability after *Gas1* deletion.** (A-C) In control, GAS1[+] cells (A, green) are in the stroma (s) and muscle (m) at E18.5. CDON (B, red) is expressed in the lower half of basal epithelium and muscle (arrow), while BOC (C, green) is expressed in the basal epithelium, stroma (s) and muscle (arrow). BOC expression in the basal epithelium (C, inset, K5, red) is predominant in the lower half (arrowheads). (D-F) In the *Gas1* mutant, the expression of GAS1 (D) is eliminated, while levels of CDON (E) and BOC (F) are maintained in the muscle (arrows), epithelium and/or stroma. (G-I) Muscle-specific *Gas1* deletion (*cGas1KO-mus*) is confirmed by GAS1 staining, which shows expression only in the stroma. Similar to the mutant, both CDON (H) and BOC (I) expression are maintained in the muscle (arrows), epithelium and/or stroma.

muscle, suggest redundant or independent regulatory roles for these co-receptors in the embryonic tongue.

### Dispensable roles for *Gas1* in the fungiform papilla taste bud primordium

Constitutive deletion of *Gas1* led to the elimination of CDON[+] cells in the immature taste bud (Fig. 3I′). However, absence of CDON along with the loss of *Gas1* from the taste bud did not affect SHH expression or paracrine HH signaling in the fungiform papilla (Fig. 3J-O), implying that GAS1 and CDON may serve HH-independent functions in this context. The dispensable role of *Gas1* in creating the SHH gradient in the forebrain is noted at E8.5 (Marczenke et al., 2021). Recently, HH-independent *Gas1* function has also been observed during kidney morphogenesis (Franks and Allen, 2024), likely due to its ability to interact with membrane receptors of Notch pathway (Marczenke et al., 2021). Given that *Gas1* deletion did not cell autonomously affect taste bud number or size at E18.5, it is possible that taste bud progenitor cells in fungiform papilla (Liu et al., 2013; Okubo et al., 2009) or innervation-mediated morphogenesis of taste bud (Barlow, 2022; Mistretta et al., 1999; Mistretta and Liu, 2006), or both, may compensate for the loss of *Gas1* in maintaining taste bud.

*Gas1* is an axon guidance receptor in the enteric nervous system (Jin et al., 2015). *Gas1* also interacts with Ret (Biau et al., 2013;

Cabrera et al., 2006; Franks and Allen, 2024), and Ret[+] nerve fibers are located outside the fungiform papillae taste buds in adults (Donnelly et al., 2018), implying that *Gas1*[+] taste cells may contribute to nerve patterning. On the other hand, *Gas1* is absent in both adult geniculate and trigeminal ganglia (Kumari et al., 2024), and if this is also true for embryonic ganglia, *Gas1* deletion will not affect nerve afferents to the tongue or taste bud. Our data show that *Gas1* loss alters neither the volume of nor innervation pattern to embryonic tongue, fungiform papillae or taste buds. Lingual innervation is not complete and established until postnatal stages (Huang et al., 2015; Krimm and Hill, 1998), so it is possible that *Gas1* may contribute to innervation surrounding fungiform papillae taste buds at postnatal stages.

### Regionalized alteration of K5[+] basal cells in *Gas1* mutant

While *Gas1* global deletion does not visibly alter the morphology or the Ki67[+] basal cell proliferation of fungiform and filiform papillae, K5 analyses demonstrate disruptions in the undifferentiated lingual epithelium cells at E18.5, specifically in the lower half of the basal layer (Figs 3I and 10B). Previous studies have suggested that other keratins can compensate in K5 mutants, resulting in normal epithelialization (Coulombe et al., 1991). However, these compensatory keratins remain fragile towards frictional trauma

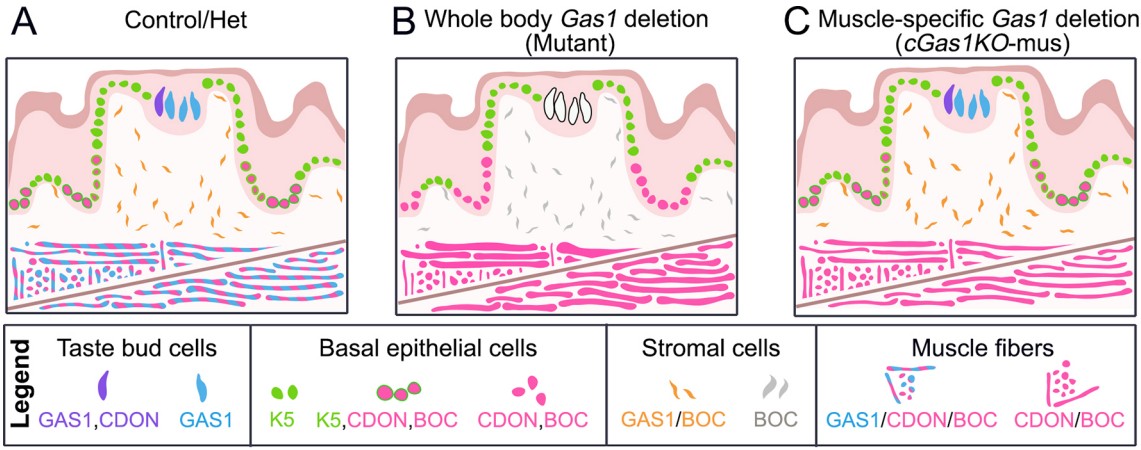

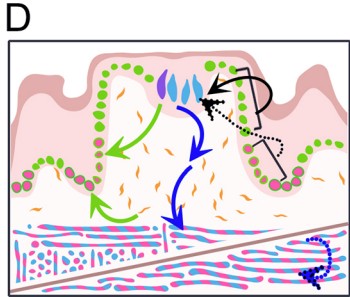

**Fig. 10. Summary and proposed models of *Gas1* functions in tongue.** (A-C) Summary diagrams of HH co-receptors (GAS1, CDON and BOC) expression in taste bud, epithelium, stroma and muscle in control/het (A), mutant (B) and *cGas1KO-mus* (C). K5 (green) expression in basal epithelium (A) is lost in the CDON$^+$ and/or BOC$^+$ cells in mutant (B). Additionally, the intrinsic muscle fiber size and count are altered in mutant (B). Tongue tissues are comparable between control (A) and *cGas1KO-mus* (C). (D) In a proposed mechanism of action, *Gas1$^+$* tissues maintain K5 expression in the lower half of fungiform papilla basal epithelium (green arrows) and are essential for the development of muscle fibers (blue arrows). The cell-intrinsic roles of *Gas1* (blue dotted arrow) are probably redundant with CDON and BOC. We propose that fungiform papilla basal epithelium is regionalized, where apical cells are essential for taste bud formation (black arrow) and basal cells are dispensable (black dotted arrows).

(Coulombe and Lee, 2012). We have not studied other keratins and cannot rule out the possibility of redundant or compensatory keratins. The data further indicate that *Gas1* haplodeficiency or muscle-specific deletion does not lead to any alterations in the epithelium (Figs 3H, 7D and 10A,C). Overall, *Gas1* may play a limited role in lingual papillae development.

Notably, K5 elimination occurs only in CDON$^+$ and/or BOC$^+$ cells in mutant (Figs 3I′ and 10B), suggesting that, in the absence of *Gas1*, CDON and/or BOC cannot maintain K5 expression. K5$^+$ proliferative basal cells give rise to fungiform papillae and taste buds (Okubo et al., 2009). Our study indicates that lingual basal cell proliferation is retained in *Gas1* mutants even in the absence of K5$^+$ cells in the lower half. We recently highlighted the regionalization of the fungiform papilla basal epithelium and suggested that the apical fungiform papilla is vital for taste bud maintenance (Ermilov et al., 2016; Kumari et al., 2017, 2024, 2018) (Fig. 10D, black arrow). Our present study further emphasizes the regionalized nature of the lingual epithelium, where *Gas1* deletion specifically eliminated undifferentiated K5$^+$ cells only in the basal half of lingual papillae (Fig. 10B). Given that fungiform papillae and taste buds are not altered in *Gas1* mutants, it might be possible that the basal half of lingual papillae is dispensable for taste bud cell differentiation (Fig. 10D, black dotted arrow).

### *Gas1* global loss severely affects fetal myogenesis
Our data suggest that *Gas1* is dispensable for lingual myoblast migration and muscle patterning during embryogenic myogenesis

(Figs 1A and 2A). However, it appears to be crucial for muscle cell proliferation, differentiation and maturation (Fig. 5H), and for intrinsic muscle fibers number and size during fetal myogenesis (Figs 6A′-D″ and 10B). This is similar to the role of Wnt signaling in limb muscle development, where it regulates progenitor cell numbers and myofiber quantity during fetal myogenesis but is not required for embryonic myogenesis (Hutcheson et al., 2009). TGFβ and/or BMP signaling in the *in vitro* limb muscles have also been shown to inhibit the differentiation of fetal myoblasts and satellite cells, but do not affect the embryonic myoblasts (Cusella-De Angelis et al., 1994). TGFβ signaling plays a crucial role in myogenic differentiation and myoblast fusion in the tongue at E14.5. However, its function beyond this stage has not yet been studied (Han et al., 2012). *Gas1* in postnatal limb muscles reportedly decreases proliferation and stem cell numbers (Li et al., 2019). Our findings support this in the embryonic tongue, as loss of embryonic *Gas1* increases proliferation and myoblast numbers.

Our studies reveal the differential regulation of various intrinsic lingual muscle types (Fig. 6), raising questions about the specificity of their developmental pathways. For example, in chick models, lingual muscle cells originate from a mixture of myogenic cells derived from different somites (Huang et al., 1999). Tongue muscles also have a mixed origin in mouse, deriving primarily from the occipital somites (Yamane, 2005), with contributions from the cranial mesoderm (Czajkowski et al., 2014). Whether transverse muscle fibers, which remain unaffected in mutant, have a distinct lineage or are subject to unique regulatory mechanisms remains unexplored.

### *Gas1+* taste bud or stromal cells regulate lingual epithelium and intrinsic muscles

Unlike *Gas1* mutant (Fig. 3I), muscle-specific deletion of *Gas1* did not alter K5 expression in the lingual epithelium (Figs 7D and 10C). Fungiform papilla basal epithelium can be supported by cranial neural crest cell (CNCC)-derived stroma (Ishan et al., 2023; Liu et al., 2004). Alternatively, paracrine signaling from the taste bud supports basal epithelium (Liu et al., 2013). Thus, we propose that *Gas1* within the fungiform papilla taste bud or stroma contributes to maintaining K5 expression in the basal layer (Fig. 10D, green arrows). Future studies with epithelium (K5) or CNCC-derived stroma (Wnt) (Liu et al., 2012) Cre models can confirm this hypothesis. Notably, in mutant, both CDON and BOC are retained in epithelium and/or stroma (Fig. 9E,F) but are unable to compensate for *Gas1* loss, suggesting independent roles of *Gas1* in these compartments.

The intricate coordination of signaling pathways from ectoderm-derived anterior tongue epithelium, CNCC-derived stromal cells or mesoderm-derived muscle progenitors is crucial for proper patterning and development of tongue musculature (Han et al., 2012; Hosokawa et al., 2010; Coren et al., 2024; Parada et al., 2012). For example, Wnt signaling produced by the lingual epithelium regulates muscle progenitor cell proliferation through Pax7 by acting upstream of Notch signaling (Zhu et al., 2017). Noncanonical TGFβ signaling and BMP4 signal in CNCC derivatives plays a crucial role in promoting myoblast proliferation and differentiation (Zhang et al., 2021; Iwata et al., 2013). Additionally, reduced levels of Pax3+ myoblast results in reduced and disorganized tongue musculature (Zhou et al., 2008). GAS1 has emerged as a potential mediator of crosstalk between Notch (Marczenke et al., 2021; Place et al., 2022), Wnt (King et al., 2024; Lee et al., 2001; Ren et al., 2016; Seppala et al., 2022) or Pax7 pathways (Li et al., 2019; Martinelli and Fan, 2007) in many tissues, but such interactions remain unexplored in tongue.

*Gas1* global deletion did not affect the lingual muscle formation and patterning at E12.5, but the subsequent development was significantly affected (Figs 2A and 10B). Thus, to understand the cell-intrinsic role of Gas1, the deletion was restricted to muscle-specific ACTA1+ cells between E12.5 and E18.5. We observed that there were no significant changes in the muscles in *cGas1KO-mus* (Figs 7, 8 and 10C). This suggests that *Gas1* within the taste bud or stroma, or both, might be essential for transducing signaling information from the epithelium and/or stroma to muscles (Fig. 10D, blue arrows), while any cell-intrinsic role of *Gas1* is dispensable (Fig. 10D, blue dotted arrow).

Additionally, it is possible that the roles of *Gas1* in muscles are redundant with other HH co-receptors, CDON and BOC. Further investigation with combined deletion of HH co-receptors is needed to delineate whether GAS1 interacts with other co-receptors in the embryonic tongue.

Overall, our study demonstrates the regulation of tongue fetal myogenesis and intrinsic muscles by the HH co-receptor *Gas1*. We have recently shown the expression of *Gas1* in postnatal tongue muscles (Kumari et al., 2024) and propose its crucial role in postnatal tongue muscle growth. Tongue muscles contribute to tongue movement and shape change (Cobourne et al., 2019), to food intake, speech and communication (Hiiemae and Palmer, 2003), and to swallowing (Le Révérend et al., 2014). Our study encourages assessing *Gas1* in lingual extrinsic muscles as well as in pediatric speech- and feeding-defect studies for its potential roles in early diagnosis and intervention.

### Study limitations

To investigate HH co-receptor expression, we employed a limited set of antibodies: one each for GAS1 and BOC, and two for CDON.

Given that the expression patterns of these co-receptors in the embryonic tongue are poorly characterized, our study provides initial insights into their localization. Both CDON antibodies yielded consistent staining patterns, suggesting specificity, and the GAS1 antibody expression closely matched *Gas1* gene expression data. Additionally, BOC staining was reproducible across multiple experiments, with no evidence of non-specific labeling. While our conclusions are based on a limited number of validated antibodies, the consistency of our findings supports the reliability of the observed expression patterns. Nonetheless, future studies using additional tools, such as reporter lines or conditional alleles for CDON and BOC, will be valuable for further validating and expanding upon these observations.

Myoblast migration into the tongue begins around E11.5, and muscle migration and patterning (embryonic myogenesis) are complete by E12.5. At E12.5, muscle fibers appeared unaltered in *Gas1* mutants, although subsequent development was significantly affected. Therefore, we initiated muscle-specific deletion at E12.5 and found no essential cell-intrinsic role for *Gas1* in fetal myogenesis. This study was not intended to add insight into the role of *Gas1* between E10.5 and E12.5 (embryogenic myogenesis). However, future studies using myoblast-specific Cre drivers (Pax3-Cre or Myf5-Cre) could determine whether early muscle precursors are regulated by *Gas1* during embryonic myogenesis or through a cell-intrinsic mechanism.

## MATERIALS AND METHODS
### Animals

All animal use and care procedures were performed in accordance with the guidelines of the National Institutes of Health and approved protocols of the Rowan-Virtua School of Osteopathic Medicine Institutional Animal Care and Use Committee. The following mouse models were used:

1. *Gas1* reporter model. We utilized heterozygote (*Gas1lacZ/+*) and control (*Gas1+/+* littermates) for *Gas1* expression studies. Tongues were collected at embryonic day (E) 12.5, E14.5, E16.5 and E18.5.
2. *Gas1* global deletion. *Gas1lacZ/+* reporter mice were bred together to generate homozygous knock-in mutant (*Gas1lacZ/lacZ*). Littermates *Gas1+/+* and *Gas1lacZ/+* were used as control and het, respectively. Tissues were collected at E12.5, E14.5, E16.5 and E18.5.
3. Muscle-specific *Gas1* deletion (*cGas1KO-mus*). An inducible *Gas1* deletion model was generated by breeding *Gas1fl/fl* mice with muscle-specific *ACTA1CreER* mice (muscle specificity of *ACTA1* is confirmed by *ACTA1CreER;RFP* mouse). Littermates negative for *Gas1fl/fl* and/or *ACTA1CreER* were used as controls (TAM control). *ACTA1CreER;Gas1fl/fl* pregnant mice were gavaged with tamoxifen (TAM) dissolved in corn oil at a dose of 100 mg/kg at E12.5, and embryonic tongues were collected at E18.5.
4. Hedgehog (HH) reporter model. *Gas1lacZ/+;Gli1GFP/+* mice were bred with *Gas1lacZ/+* to map HH-responding GLI1+ cells in control, het and mutant.

*Gas1lacZ/+* reporter mice (Martinelli and Fan, 2007) and *Gas1fl/fl* mice (Jin et al., 2015) were generous gifts from Dr Cheng Ming Fan (Carnegie Institution for Science, Baltimore, MD, USA). *Gli1GFP/+* reporter mice (Brownell et al., 2011) were a generous gift from Dr Alexandra Joyner (Sloan Kettering Institute Cancer Center, New York, NY, USA). *ACTA1CreER* (031934) and RFP (007914) mice were obtained from the Jackson Laboratory. PCR was used to confirm embryo genotypes. We used a minimum of three animal tongues per group.

### Embryonic tongue collection and processing

Male and female mice were placed in the same cage in the afternoon. Vaginal plugs were tracked every morning to confirm mating between mice. The day the plug was observed was denoted a E0.5. Mouse weight was also tracked to confirm pregnancy. On the specified day, the pregnant female was euthanized with $CO_2$ overdose and cervical dislocation. A C-section was

performed to collect embryos. Embryos were euthanized with decapitation. Whole heads were fixed for 30 min (E12.5 *Gas1^lacZ*), 2 h (E14.5-E18.5 *Gas1^lacZ*) or 5 h (E18.5 *ACTA1^CreER*;*Gas1^fl/fl*) in 4% paraformaldehyde in phosphate-buffered saline (PBS) at 4°C. Tongues were dissected from the mandible, washed with PBS, cryoprotected overnight with 30% sucrose in PBS and then embedded in O.C.T. compound. Serial sections were cut at 10 µm and mounted onto charged glass slides. For the E12.5 stage, the whole head was processed to cut into sagittal or coronal sections. For intrinsic muscle analysis, E18.5 tongues were sectioned in a coronal, sagittal or horizontal plane to analyze longitudinal (superior, SLm; inferior, ILm), transverse (Tm) and vertical (Vm) muscle fibers, respectively. Sagittal sections were used to analyze lingual epithelium, stroma and innervation.

### Staining
Tongue sections were stained with Hematoxylin and Eosin (H&E) or X-gal, or immunostained using antibodies, as described previously (Kumari et al., 2015, 2017, 2024, 2022, 2018). Briefly, for H&E staining, tongue sections were air dried, dehydrated in ethanol, defatted in xylene, rehydrated in ethanol and stained in Gill's III Hematoxylin for 30 s and Eosin for 30 s. Tongue sections were incubated for 1 h (E12.5) or 2 h (E14.5-E18.5) at 37°C with X-Gal solution for X-Gal staining. For immunostaining, tongue sections were blocked for 30 min at room temperature in 10% normal donkey serum in PBS-X (0.3% Triton-X 100 in PBS, pH 7.4) and incubated overnight at 4°C with primary antibodies. The following day, sections were washed in PBS, incubated with the corresponding secondary antibody for 1-2 h at room temperature in the dark, air dried and mounted with a solution containing DAPI. The list of primary and secondary antibodies is provided in Table S1. We also performed Masson's trichrome staining in sagittal tongue sections, as per standard protocol (Kiernan, 2015). Briefly, air-dried tongue sections were re-fixed in Bouin's solution for 1 h and sequentially stained in Weigert's iron Hematoxylin (5 min), Biebrich scarlet-acid fuchsin (15 min), phosphomolybdic-phosphotungstic acid (10 min), aniline blue (10 min) and 1% acetic acid (1 min) followed by dehydration, clearing and mounting. Collagen was detected as blue and cytoplasm as red.

### Imaging
Whole head or tongue was imaged using a VWR stereo zoom trinocular microscope. Tongue section imaging was performed using automated fluorescence microscopes Keyence BZX-710 and 810. BZ-X Analyzer Z-stacking was used to process images. Figures were assembled with Adobe Photoshop. Adjustments to brightness and contrast were made across all images of a figure, if required.

### Qualitative data analyses
Between control, het and mutant groups, images of tongue sections labelled with an array of staining were used to compare the epithelium (H&E, X-gal and K5 immunostaining), stroma (Masson's trichrome staining and vimentin immunostaining), innervation (β-tubulin, NF and P2X3 immunostaining), muscle cell fusion (desmin immunostaining) and HH pathway components (SHH, GLI1-GFP, GAS1, CDON and BOC immunostaining). Masson's trichrome and K5, GAS1, CDON and BOC-immunostained images were also descriptively analyzed in TAM control and *cGas1KO-mus*. A minimum of eight images per animal were used.

### Quantitative data analyses
#### Phenotype
Using NIH ImageJ software for whole head/tongue images, we measured head length (from the nose tip to the head base), tongue width (at the middle-most part of the tongue) and jaw angle (angle at which the sides of the jaw converge at the tip).

#### Tongue
Tongue length and height were measured using tongue sections and NIH ImageJ software. Length was measured from the tongue tip to the base, as indicated by H&E staining (Fig. 2E, control, dotted line). Height measurements were taken at three different regions of the tongue: tip, body and posterior, as indicated by H&E staining (Fig. 2E, control, arrows). The height was measured from the dorsum to the ventral side (dotted line). The

tip is defined as the point where the ventral epithelium becomes smooth, transitioning out of the lingual papillae. The posterior is located at the point where the dorsum becomes smooth. The body height was averaged from three locations in the middle of the tongue. Both H&E and X-gal stained tongue sections were used. A minimum of six non-contiguous images per tongue, around the middle to avoid edges, were analyzed.

#### Fungiform papilla
The fungiform papilla (Kumari et al., 2015, 2017, 2018) was counted in 600 µm sections of the tongue, away from the rounded edges, and represented as fungiform papilla density per 600 µm of tongue. In the same slides, serial sections of fungiform papilla were analyzed and categorized as typical morphology (rectangular, thin keratinized apex, single taste bud, Fig. 3A) or atypical morphology (altered shape/keratinization with no taste bud). Data are represented as the percentage of typical fungiform papilla. X-Gal- or K8 immunostaining was used to analyze the area of fungiform papilla taste bud with NIH ImageJ software. Ki67$^+$ proliferating cells in fungiform papilla basal epithelium were counted and represented as the number of Ki67$^+$ cells per fungiform papilla. A total of 22-28 taste buds (for taste bud area) and 18-29 fungiform papilla (for Ki67 count) per group (*n*=3) were quantified.

#### Stroma
At E18.5, control, het and mutant embryos were euthanized, and lingual stromal tissues were collected to isolate RNA (Qiagen RNeasy Plus Micro Kit 74034), followed by cDNA synthesis (Thermo Fisher High-Capacity cDNA Reverse Transcription Kit 4374966) for vimentin qPCR (Agilent Aria Mx) using primers: Fwd, GTG ATG TGC GCC AGC AGT ATG; Rev, TGC AGG CGG CCA ATA GTG. The ddCT method (Livak and Schmittgen, 2001) was employed to quantify changes in gene fold expression.

#### Muscle
In sagittal tongue sections, immunostaining was used to label proliferating cells (Ki67), myoblasts (Pax7), differentiating fibers (myogenin) and mature fibers (myosin heavy chain, MyHC, MyHC-embryonic, MyHC-neonatal and MyHC-slow). All positive signals were counted in non-overlapping and non-contiguous images using NIH ImageJ software. Proliferating cells are presented as the percentage of Ki67$^+$ to DAPI$^+$ cells. Pax7, myogenin and MyHC data are presented as counts per image per tongue. MyHC types (MyHC-embryonic, MyHC-slow and MyHC-neonatal) data are expressed as the percentage of control per image per tongue. Each animal had 4 to 16 images.

Additionally, α-bungarotoxin staining (Thermo Fisher T1175) was used to mark acetylcholine receptors (AChRs). The number of AChRs is presented as the AChR count per 600 µm$^2$ of tongue. The area of AChR is measured only from enfaced AChR (Bahia El Idrissi et al., 2016) and presented per tongue. Co-expression of presynaptic nerve (SNAP25) and AChR was used to measure the percentage of innervated AChRs. AChRs were classified as 'vacant' if there was no overlap between the SNAP25 and α-bungarotoxin signals or 'full' if there was >80% overlap. Data are presented as the percentage of vacant or full AChRs per tongue. Each tongue had a minimum of 15 images. In addition, we quantified amount of percentage overlap between AChR and SNAP25, and expressed it as a fraction of the postsynaptic CSA using NIH ImageJ software (Jones et al., 2016). The number of axonal inputs per AChR is presented per tongue. Each tongue had a minimum of 10 images.

Laminin immunostaining was used to outline muscle fiber for counting and measuring cross-sectional area (CSA) using SMASH MATLAB version R2014a (Smith and Barton, 2014). Tongues were sectioned in different planes to study the distinct intrinsic muscles: coronal (SLm and ILm), sagittal (Tm) and horizontal (Vm). Quantifications were made in four or more non-overlapping and non-contiguous tongue images from the indicated regions (Fig. 6A-D, white boxes). Muscle fiber counts are summed and presented as the count per tongue. The CSA was averaged per tongue, and the relative frequency of each fiber type is represented by the number of fibers in each CSA interval per tongue.

### Statistics
Statistical analyses were performed using SPSS Statistics software version 29. Data in figures are presented as the median±1.5 times the

interquartile range (i.e. maximum or minimum). Graphs were generated using R/RStudio. A one-way ANOVA was used to compare differences between control, het and mutant groups, followed by Tukey's post-hoc test. A two-way ANOVA was used to assess the interactions between genotype (control, het and mutant) and two factors, followed by Tukey's post-hoc test. Unpaired *t*-tests were used to compare TAM control and *cGas1KO-mus* groups. Significance was set at $P \leq 0.05$ (*$P \leq 0.05$, **$P \leq 0.01$, ***$P \leq 0.001$).

## Acknowledgements

We acknowledge Drs Cheng Ming Fan and Alexandra Joyner for generously sharing the mouse models. We gratefully acknowledge Rowan-Virtua SOM medical student Ashlyn P. McClelland, funded by the Summer Medical Research Fellowship program, and Rowan University undergraduate student Aysenur Sen for their valuable assistance with staining and imaging during the paper revision. We also appreciate the support from the Rowan-Virtua SOM Neuroscience Department's Histology, Common Equipment and Imaging Core Facilities, and Dr Renee M. Demarest for her assistance with H&E and Masson's Trichrome staining.

## Competing interests

The authors declare no competing or financial interests.

## Author contributions

Conceptualization: A.K.; Data curation: G.C.A.; Formal analysis: G.C.A., S.Y.R., A.K.; Funding acquisition: A.K.; Investigation: G.C.A., S.Y.R.; Methodology: G.C.A., A.K.; Project administration: A.K.; Resources: A.K.; Supervision: A.K.; Validation: A.K.; Visualization: A.K., G.C.A.; Writing – original draft: A.K.; Writing – review & editing: G.C.A., S.Y.R., A.K.

## Funding

This work was supported by the National Institutes of Health National Institute on Deafness and Other Communication Disorders (ECR R21 1R21DC017799 to A.K.). Further support was provided by the Rowan University seed fund (A.K.) and Rowan Federal Work-Study Program (S.Y.R.). Open Access funding provided by Rowan University. Deposited in PMC for immediate release.

## Data and resource availability

The data that support the findings of this study are openly available in Figshare with the identifier: https://doi.org/10.6084/m9.figshare.28204280.v1. All relevant data and details of resources can be found within the article and its supplementary information.

## The people behind the papers

This article has an associated 'The people behind the papers' interview with some of the authors.

## Peer review history

The peer review history is available online at https://journals.biologists.com/dev/lookup/doi/10.1242/dev.204868.reviewer-comments.pdf.

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
