## [Peer Review File · Development (Cambridge, England)]

Gas1 regulates embryonic tongue muscle proliferation, differentiation and maturation via alternative pathways to Hedgehog signaling

Gabrielle C. Audu, Sally Y. Rohan and Archana Kumari

DOI: 10.1242/dev.204868

Editor: Liz Robertson

Review timeline

Original submission:	15 April 2025
Editorial decision:	23 May 2025
First revision received:	18 August 2025
Accepted:	1 September 2025

Original submission

First decision letter

MS ID#: dev.204868

MS TITLE: Gas1 regulates embryonic tongue muscle proliferation, differentiation, and maturation independent of Hedgehog signaling

AUTHORS: Gabrielle C. Audu, Sally Y. Rohan and Archana Kumari

Dear Dr Kumari,

I have now received all the referees' reports on the above manuscript, and have reached a decision. The referees' comments are appended below, or you can access them online: please go to:

As you will see, the referees express considerable interest in your work, but have some significant criticisms and recommend a substantial revision of your manuscript before we can consider publication. If you are able to revise the manuscript along the lines suggested, which may involve further experiments, I will be happy receive a revised version of the manuscript. Your revised paper will be re-reviewed by one or more of the original referees, and acceptance of your manuscript will depend on your addressing satisfactorily the reviewers' major concerns. Please also note that Development will normally permit only one round of major revision. If it would be helpful, you are welcome to contact us to discuss your revision in greater detail. Please send us a point-by-point response indicating your plans for addressing the referees' comments, and we will look over this and provide further guidance.

Please attend to all of the reviewers' comments and ensure that you clearly highlight all changes made in the revised manuscript. Please avoid using 'Tracked changes' in Word files as these are lost in PDF conversion. I should be grateful if you would also provide a point-by-point response detailing how you have dealt with the points raised by the reviewers in the 'Response to Reviewers' box. If you do not agree with any of their criticisms or suggestions please explain clearly why this is so.

Reviewer 1

Advance summary and potential significance to field

Audu and colleagues present a highly interesting and elegant study on tongue muscular and sensory development, evaluating the effects of Gas1 using multiple developmental ages and deletion strategies.

This work shows extensive effects on muscle, independent of the effects on hedgehog developmental signalling. As normal tongue development is essential for neonatal survival, this study provides a great resource for future studies on the intricacies of taste and lingual development. Previous efforts have focused on "master" developmental regulators, while this evaluation illuminates the nuance and timings of regulators at later developmental stages. Evaluation of all 4 intrinsic muscles and their NMJs is a great advance.

I have some specific queries below to enhance the clarity and authority of the study.

Specific Comments

- 1 - Are the epithelial and myofiber cell precursor lineages the same in the tongue?
- 2 - NMJs are continually undergoing denervation reinnervation, perhaps the best evaluation of presynaptic invasion of the endplate is the % of overlap as expressed as a fraction of the postsynaptic cross-sectional area or volume.
- 3 - Did Gas1 deletion alter the fraction of polyneuronal NMJ innervation?
- 4 - it looks like it is possible to do absolute counts of individual tongue muscle fibres, which would be better than the notoriously variable density counts.
- 5 - Did gas1 deletion alter the relative amounts of MyHCemb, MyHCneo and MyHCslow
- 6 - The supplementary figures are important, and quite beautiful in their own way and could be incorporated into the main body.
- 7 - Did Gas1 deletion, with its major effects on tongue musculature have any effect on hypoglossal MN numbers, which are undergoing marked programmed neuronal death during this period?

Reviewer 2

Advance summary and potential significance to field

The developmental role of Gas1 in tongue muscle has not been previously explored, and the phenotypes presented in this manuscript are interesting. While I am not a specialist in muscle biology and therefore may not be best positioned to evaluate the full significance of the findings, the authors' characterization and quantification of the observed phenotypes are well-executed and informative. However, the claim that these phenotypes occur independently of Hedgehog signaling is not sufficiently supported by the current data. Additional evidence or a more cautious interpretation focusing on the role of Gas1 in muscle development would strengthen the manuscript.

Comments for the author

Several claims in the manuscript would benefit from more cautious interpretation, as they are not fully supported by the current data.

First, the conclusion that Hedgehog signaling is not involved in muscle development is not directly substantiated by experimental evidence. While the Gas1 conditional knockout does not exhibit phenotypes, this alone does not rule out Hedgehog pathway involvement. Gas1 is one of several co-receptors, and Hedgehog ligands can still activate signaling through direct interaction with Ptch1. Therefore, the data do not exclude a potential role for Hedgehog signaling, and this interpretation should be presented more cautiously throughout the manuscript and not be used in the title.

Second, the suggestion that Gas1 acts in a non-cell-autonomous manner across compartments such as the epithelium and stroma is not directly supported by experimental data. This claim appears speculative and would be strengthened by focusing on the more directly supported cell-intrinsic roles of Gas1 in muscle development. There are also inconsistencies in the manuscript regarding the proposed non-cell-autonomous role of Gas1 in tongue myogenesis. Although the authors suggest a non-cell-autonomous function, they also state (line 309) that Gas1 protein expression fully aligns

with its gene expression. This alignment supports a cell-autonomous role, which contradicts their main claim. Furthermore, Figure 1C'-F' shows minimal LacZ staining in the stroma, which is inconsistent with the Results section (line 102) that reports stromal expression."

Third, the claim that the "basal half of the fungiform papilla is dispensable for taste bud cell differentiation" (line 379-381) is not supported by experimental evidence and should not be proposed in the Discussion. The authors did not assess whether basal cells lacking K5 expression exhibit any differentiation defects. Moreover, the observation that Ki67+ basal cell proliferation remains unaffected suggests that Gas1 is not required for their proliferation and should be considered in Discussion.

Additionally, the ACTA1-driven muscle-specific conditional knockout did not recapitulate the phenotype observed in the global knockout, which included disruptions in fetal myogenesis and increased proliferation of muscle progenitor cells. This discrepancy may stem from the relatively late onset of ACTA1 expression, which occurs after progenitor cells have already committed to differentiation. The manuscript would benefit from a discussion of the timing and specificity of Gas1 deletion in this model, as well as the potential utility of earlier-acting Cre drivers to more accurately assess Gas1 function during myogenesis in the tongue.

In summary, while the study provides interesting data, the conclusions rely heavily on negative results from a conditional model that may not effectively capture early roles of Gas1. A more delicate and focused interpretation, along with acknowledgment of the limitations in experimental design, would strengthen the overall impact of the manuscript.

- Additional minor comments.

* Figure 1 C'-F' staining in stroma not visible

* Figure 3H' the CDON staining in TB is not visible

* Figure 4B the collagen staining in Mutant is not visible

* Line 313-316: The sentence "postnatal expression decline and stroma and elimination in muscle" is not supported by data. Additionally, it does not logically support the claim that follows. Please revise or remove.

* Line 337: The phrase 'deletion of both Gas1 and Cdon' is misleading, as it implies that double knockout animals were generated and analyzed. However, the manuscript does not present data from such a model. The authors should revise this wording to more accurately reflect the experimental approach used.

* Line 353, GG and TG are not defined

* Several figure references are missing in the Discussion—for example, in lines 327 and 337. The authors should carefully review the manuscript to ensure that all claims are appropriately supported by references to the corresponding Results and Figures.

First revision

Author response to reviewers' comments

Point-by-point responses to reviewer comments:

Reviewer 1: SUMMARY OF THE ADVANCE MADE IN THIS PAPER AND ITS POTENTIAL SIGNIFICANCE TO THE FIELD

Audu and colleagues present a highly interesting and elegant study on tongue muscular and sensory development, evaluating the effects of Gas1 using multiple developmental ages and deletion strategies.

This work shows extensive effects on muscle, independent of the effects on hedgehog developmental signalling. As normal tongue development is essential for neonatal survival, this study provides a great resource for future studies on the intricacies of taste and lingual

development. Previous efforts have focused on "master" developmental regulators, while this evaluation illuminates the nuance and timings of regulators at later developmental stages.

Evaluation of all 4 intrinsic muscles and their NMJs is a great advance.

Response: We thank the reviewer for the positive comments.

I have some specific queries below to enhance the clarity and authority of the study. Specific Comments

1 - *Are the epithelial and myofiber cell precursor lineages the same in the tongue?*

Response: No, the epithelial and myofiber cell precursor lineages in the tongue are distinct and arise from different embryonic origins. Epithelial cell precursors of the anterior and posterior tongue originate from the oral ectoderm and endoderm, respectively. In contrast, myofiber cell precursors derive from mesoderm, specifically from occipital somites. Some of this information is now included in line # 438-440.

2 - *NMJs are continually undergoing denervation reinnervation, perhaps the best evaluation of presynaptic invasion of the endplate is the % of overlap as expressed as a fraction of the postsynaptic cross-sectional area or volume.*

Response: Thank you for the suggestion. In addition to quantifying fully innervated and vacant NMJs, we have now quantified the actual percentage overlap between presynaptic SNAP25+ fibers and postsynaptic endplates. This overlap is expressed as a fraction of the postsynaptic endplate cross-sectional area and is presented in Supplementary Figure S2C. The extent of presynaptic invasion showed no correlation with endplate cross-sectional area in any of the experimental groups. Method section, line # 612-614 and the Result section, line # 238-239 are updated accordingly.

3 - *Did Gas1 deletion alter the fraction of polyneuronal NMJ innervation?*

Response: We performed additional quantification to assess the number of axons converging onto individual NMJs. Our analysis revealed that *Gas1* deletion substantially reduced the number of axonal inputs per NMJ (Figure S2B, lines 237-238). We observed fewer than three axonal inputs, likely due to the 10 μ m section thickness or the fact that polyneuronal innervation in mouse tongue muscles peaks between E11 and E15, followed by rapid synapse elimination occurring after E15 (Yamane et al., 2000; <https://doi.org/10.2108/zsj.17.935>).

4 - *it looks like it is possible to do absolute counts of individual tongue muscle fibres, which would be better than the notoriously variable density counts.*

Response: Thank you for the comment. We have reprocessed the data to determine absolute counts of individual tongue muscle fibers. The new graphs Figure 6A'-D' represent absolute counts.

5 - *Did gas1 deletion alter the relative amounts of MyHCemb, MyHCneo and MyHCslow*

Response: Thank you for this insightful comment. We have done additional experiments to include data for MyHCemb (Figure 5F,F'), MyHCslow (Figure S1J,J') and MyHCneo (Figure S1K,K'). As with overall myosin heavy chain counts, MyHCemb was significantly reduced in mutants compared to control and het (line # 223-224). MyHCslow was also substantially reduced in mutant, whereas MyHCneo remained unchanged (line # 225-226). Method section (line # 601-603) and supplementary Table 1 are updated accordingly.

6 - *The supplementary figures are important, and quite beautiful in their own way and could be incorporated into the main body.*

Response: We thank the reviewer for this comment. Constraints on figure size and space within the manuscript limit our ability to incorporate all supplementary figures into the main body. However, we have integrated S1B, S1D, S1E, S1L, S2D, S2E and S2F by revising existing figures or creating new ones.

7 - *Did Gas1 deletion, with its major effects on tongue musculature have any effect on hypoglossal MN numbers, which are undergoing marked programmed neuronal death during this period?*

Response: Our current study focused on the peripheral effects of *Gas1* deletion within the tongue tissue compartments. Thus, we did not examine hypoglossal motor neuron numbers

within the hypoglossal nucleus (nXII) in the brainstem in this study.

Reviewer 2: SUMMARY OF THE ADVANCE MADE IN THIS PAPER AND ITS POTENTIAL SIGNIFICANCE TO THE FIELD

The developmental role of Gas1 in tongue muscle has not been previously explored, and the phenotypes presented in this manuscript are interesting. While I am not a specialist in muscle biology and therefore may not be best positioned to evaluate the full significance of the findings, the authors' characterization and quantification of the observed phenotypes are well-executed and informative. However, the claim that these phenotypes occur independently of Hedgehog signaling is not sufficiently supported by the current data. Additional evidence or a more cautious interpretation focusing on the role of Gas1 in muscle development would strengthen the manuscript.

Response: We thank the reviewer for this thoughtful and constructive feedback. We would like to clarify that our intention was to suggest that the role of *Gas1* in tongue muscle development may involve mechanisms that are independent of canonical Hedgehog (HH) signaling, rather than claiming that the observed phenotypes occur entirely independently of HH signaling. In response to the reviewer's comment, we have revised the Discussion section to present a more cautious and nuanced interpretation of our findings, with clearer language regarding the potential relationship between *Gas1* and HH signaling pathways.

SUGGESTIONS TO AUTHORS

Several claims in the manuscript would benefit from more cautious interpretation, as they are not fully supported by the current data.

First, the conclusion that Hedgehog signaling is not involved in muscle development is not directly substantiated by experimental evidence. While the Gas1 conditional knockout does not exhibit phenotypes, this alone does not rule out Hedgehog pathway involvement. Gas1 is one of several co-receptors, and Hedgehog ligands can still activate signaling through direct interaction with Ptch1. Therefore, the data do not exclude a potential role for Hedgehog signaling, and this interpretation should be presented more cautiously throughout the manuscript and not be used in the title.

Response: As noted above, our intent was to suggest that *Gas1* may act through mechanisms independent of canonical HH signaling, not that the observed phenotypes are entirely HH-independent. The absence of phenotypes in *Gas1* conditional knockout indicates that cell-intrinsic functions in muscles can be compensated. In response to the reviewer's comment, we have revised the manuscript and the title.

Revised title: Gas1 regulates embryonic tongue muscle proliferation, differentiation, and maturation via alternative pathways to Hedgehog signaling

Second, the suggestion that Gas1 acts in a non-cell-autonomous manner across compartments such as the epithelium and stroma is not directly supported by experimental data. This claim appears speculative and would be strengthened by focusing on the more directly supported cell-intrinsic roles of Gas1 in muscle development. There are also inconsistencies in the manuscript regarding the proposed non-cell-autonomous role of Gas1 in tongue myogenesis. Although the authors suggest a non-cell-autonomous function, they also state (line 309) that Gas1 protein expression fully aligns with its gene expression. This alignment supports a cell-autonomous role, which contradicts their main claim. Furthermore, Figure 1C'-F' shows minimal LacZ staining in the stroma, which is inconsistent with the Results section (line 102) that reports stromal expression."

Response: Our primary aim was to acknowledge that the *Gas1*⁺ epithelium and stromal cells supply paracrine cues essential for tongue myogenesis and to raise the possibility that *Gas1* loss in these compartments could secondarily influence muscle maturation. Thus, to avoid any misinterpretation, "non-cell-autonomous" mechanism has been replaced with "cross-compartmental effects" in the Abstract (line # 39) and Summary statement (line # 43). We have adjusted the images in Figure 1D'-G' (old 1C'-F') for enhanced clarity. We have also noted in the Results that the *Gas1* stromal expression increases over time (line # 127-128).

Third, the claim that the "basal half of the fungiform papilla is dispensable for taste bud cell differentiation" (line 379-381) is not supported by experimental evidence and should not be proposed in the Discussion. The authors did not assess whether basal cells lacking K5 expression exhibit any differentiation defects. Moreover, the observation that Ki67⁺ basal cell

proliferation remains unaffected suggests that Gas1 is not required for their proliferation and should be considered in Discussion.

Response: We have revised the discussion title to “Regionalized alteration of K5+ basal cells in *Gas1* mutant” (line #382), along with the associated text in the following paragraphs, to clarify that *Gas1* is not required for lingual papillae proliferation, and that K5+ basal cells alteration are regionalized rather than differentiation defects.

Additionally, the ACTA1-driven muscle-specific conditional knockout did not recapitulate the phenotype observed in the global knockout, which included disruptions in fetal myogenesis and increased proliferation of muscle progenitor cells. This discrepancy may stem from the relatively late onset of ACTA1 expression, which occurs after progenitor cells have already committed to differentiation. The manuscript would benefit from a discussion of the timing and specificity of Gas1 deletion in this model, as well as the potential utility of earlier-acting Cre drivers to more accurately assess Gas1 function during myogenesis in the tongue.

Response: We thank the reviewer for this thoughtful comment. We have added two schematics (Figure 1A, 7A) and revised the text (line # 48-54) to be clearer about the timing and specificity of *Gas1* deletion models. As shown in Figure 1A, tongue development begins around E10.5, with muscle progenitor migration and organization completed by E12.5.

These early events appear unaffected in the *Gas1* mutant, as muscles are seen at E12.5 (Figure 2A). Based on this, we focused our analysis on the muscle-specific roles of *Gas1* after E12.5 (fetal myogenesis). This rationale is added in the Results (line # 278-281). We confirmed that ACTA1 expression begins at E12.5, using an RFP reporter line (Fig 7B).

This study was not intended to add insight into the role of *Gas1* between E10.5 and E12.5 (embryonic myogenesis). This limitation is now acknowledged in the manuscript, along with a suggestion to use myoblast-specific Cre drivers such as Pax3-Cre or Myf5-Cre in future studies (line # 480-487).

In summary, while the study provides interesting data, the conclusions rely heavily on negative results from a conditional model that may not effectively capture early roles of Gas1. A more delicate and focused interpretation, along with acknowledgment of the limitations in experimental design, would strengthen the overall impact of the manuscript.

Response: We have extensively revised the manuscript, including discussion to better emphasize our focus on studying fetal myogenesis. Additionally, we now acknowledge as a limitation of the study that, although our data do not indicate alterations in embryonic myogenesis in the *Gas1* mutant, we did not examine this using early-stage, muscle-specific Cre drivers such as *Pax3Cre* or *Myf5Cre*.

- Additional minor comments.

* *Figure 1 C'-F' staining in stroma not visible*

Response: Adjustments were made to enhance their visibility (new Figure 1D'-G').

* *Figure 3H' the CDON staining in TB is not visible*

Response: Insets and arrowheads are added in Figure 3G' and H' to indicate CDON staining in taste bud.

* *Figure 4B the collagen staining in Mutant is not visible*

Response: We have replaced the original image with one that more clearly represents the collagen staining in the mutant.

* *Line 313-316: The sentence "postnatal expression decline and stroma and elimination in muscle" is not supported by data. Additionally, it does not logically support the claim that follows. Please revise or remove.*

Response: Missed reference to this work is now included (line # 335-338). We have revised the sentence that follows.

Revised sentences: Unlike the postnatal decline in *Gas1* expression in the stroma and its elimination in muscle tissue (Kumari et al., 2024), *Gas1* expression remains robust in both compartments during embryonic tongue development. **This sustained expression pattern suggests a continuous requirement for *Gas1* during embryonic development of tongue tissues, though functional validation is needed to confirm this role.**

* *Line 337: The phrase 'deletion of both Gas1 and Cdon' is misleading, as it implies that double knockout animals were generated and analyzed. However, the manuscript does not present data from such a model. The authors should revise this wording to more accurately reflect the experimental approach used.*

Response: Revised the sentence for clarity (line # 360-361).

Revised sentence: Absence of CDON along with the loss of Gas1 from the taste bud did not affect SHH expression or paracrine HH signaling in the fungiform papilla.

* *Line 353, GG and TG are not defined*

Response: GG and TG as geniculate ganglion and trigeminal ganglion are now defined (line # 375-376).

* *Several figure references are missing in the Discussion—for example, in lines 327 and 337. The authors should carefully review the manuscript to ensure that all claims are appropriately supported by references to the corresponding Results and Figures.*

Response: Thank you for the suggestion. We have added main figures information into the discussion.

Second decision letter

MS ID#: dev.204868R1

MS TITLE: Gas1 regulates embryonic tongue muscle proliferation, differentiation, and maturation via alternative pathways to Hedgehog signaling

AUTHORS: Gabrielle C. Audu, Sally Y. Rohan and Archana Kumari

Dear Dr Kumari,

I am happy to tell you that your manuscript has been accepted for publication in Development, pending our standard publication integrity checks.

Reviewer 1

Advance summary and potential significance to field

This is a great study highlighting an important area in the context of the development of a very complicated organ

Comments for the author

Reviewer 2

Advance summary and potential significance to field

This manuscript makes an important contribution to the field of developmental biology by demonstrating an early requirement of Gas1 in tongue muscle development. The mutant phenotypes are carefully characterized, revealing a specific and unexpected role for Gas1 in one type of intrinsic tongue muscle, while other tongue muscles remain unaffected. This level of specificity is intriguing, as it points toward muscle-type dependent genetic regulation during craniofacial development. The authors have fully and thoughtfully addressed the comments and suggestions raised during the review process. The revised manuscript has been strengthened substantially.

Comments for the author

The authors may consider expanding the discussion to address potential relevance to human conditions characterized by microglossia, where early disruptions in tongue muscle development can contribute to craniofacial anomalies.